# SplitQuant: Efficient Low-Bit Quantization for Diffusion Transformers via In-Channel Dimension Splitting

## Abstract

Diffusion models currently dominate the field of image generation. However, generating high-resolution images requires larger-scale diffusion models that consume substantial computational resources and memory during inference. While post-training quantization offers a promising solution to reduce computational costs and memory usage through low-precision representations, existing approaches face significant challenges when applied to diffusion models. Unlike large language models that are memory-bound, **Di**ffusion **T**ransformers (DiT) are compute-intensive during inference. Consequently, current methods that rely on additional parameters to recover the performance of extremely low-bit quantized models achieve minimal acceleration benefits, as they introduce non-negligible computational overhead. To address these challenges, we propose **SplitQuant**, a novel approach that reduces additional computational overhead while improving low-bit quantization performance by strategically splitting the in-channel dimension of linear layers and activations. Recognizing that diffusion transformer architectures differ fundamentally from large language models, we develop a specialized optimization pipeline tailored specifically for diffusion models, which significantly enhances the generation quality of low-bit quantized models. Additionally, we implement custom-optimized CUDA kernels for SplitQuant that render the preprocessing overhead from additional parameters and quantization processes negligible, achieving single-operator performance comparable to W4A4 QGeMM across various tensor shapes. Extensive experiments on FLUX.1, PixArt-$\Sigma$, and Wan2.1 demonstrate SplitQuant's effectiveness in both image generation scenarios. Our method achieves $2.7\times$ acceleration on linear layer operators across different shapes, with SplitQuant kernels delivering performance that approaches Int4 QGeMM acceleration. The code is available at this anonymous link.

## 1 Introduction

Diffusion Transformers (DiTs) (Peebles & Xie, 2023) have emerged as the dominant architecture in image and video generation, demonstrating remarkable generative capabilities through iterative denoising processes. Building upon the inherent scalability advantages of transformer architectures (Vaswani et al., 2017), researchers are employing increasingly larger models trained on extensive high-quality datasets to develop more powerful generative systems. Moreover, the architectural convergence between language and vision models has enabled a unified paradigm that integrates generation and understanding capabilities (Hurst et al., 2024; Comanici et al., 2025; Deng et al., 2025). This convergence, combined with training on large-scale diverse datasets, has given rise to enhanced generative capabilities encompassing image editing (Kawar et al., 2023; Brooks et al., 2023; Huang et al., 2025), long-context interaction (Wu et al., 2024a; Zhang et al., 2024), cross-modal understanding (Fu et al., 2025), and multimodal generation (Wu et al., 2025; Chen et al., 2025b; Zhang et al., 2025).

However, these powerful generative capabilities come at the cost of significant computational and memory demands. Model quantization (Xiao et al., 2023; Lin et al., 2024; Shao et al., 2023; Ma et al., 2024b; Sun et al., 2024; Ashkboos et al., 2024; Liu et al., 2024) addresses these challenges by converting high-precision floating-point representations to low-precision floating-point

or fixed-point formats, thereby reducing both model size and computational overhead. Like large language models, diffusion transformers exhibit extreme activation outliers during inference (Zhao et al., 2024a; Li et al., 2024b; 2025), which lead to severe quantization errors. Unfortunately, existing quantization algorithms designed to mitigate outliers in large language models prove inadequate for diffusion models due to fundamental architectural differences and algorithmic constraints. SmoothQuant (Xiao et al., 2023) attempts to address this issue by applying channel scaling to the in-channel dimensions of linear layers and activations, effectively redistributing activation outliers to the weight space to minimize quantization errors. However, SmoothQuant demonstrates limited effectiveness when handling extreme outliers, particularly under ultra-low-bit quantization settings (e.g., W4A4), where it fails to preserve model performance. Although rotation-based methods (Ashkboos et al., 2024; Liu et al., 2024) show promise in resolving outlier issues through high-dimensional matrix transformations, they face a critical limitation in diffusion models: the timestep information injected at each inference step prevents matrix fusion, necessitating extensive online matrix computations that negate the compression and acceleration benefits of quantization. Specifically, the distributive and associative properties do not hold for matrix multiplication over the element-wise multiplication between block outputs and timestep modules (gate_msa, gate_mlp), thereby preventing forward fusion of rotation matrices, as formalized below:

$$
\begin{cases}
(X_{\text{gate\_i}} \odot X_{\text{i}}) \times R \neq X_{\text{gate\_i}} \odot (X_{\text{i}} \times R), \\
(X_{\text{gate\_i}} \odot X_{\text{i}}) \times R \neq (X_{\text{gate\_i}} \times R) \odot (X_{\text{i}} \times R).
\end{cases}
\tag{1}
$$

Where $i \in \{msa, mlp\}$, $R$ represents the rotation matrix, $\odot$ denotes element-wise matrix multiplication, and $\times$ denotes matrix multiplication. Equation 1 demonstrates that the rotation matrix $R$ cannot be fused into the linear layers of either the timestep modules or the backbone modules. The storage requirements for high-dimensional rotation matrices $R$ and their online computation during inference result in high latency and substantial memory consumption, thereby negating the quantization benefits. Consequently, the unique architectural characteristics of diffusion models preclude the effective adaptation of rotation transformation-based methods.

To address this limitation, an alternative class of methods tackles the outlier problem in Transformers by introducing minimal additional parameter computations without requiring matrix fusion. FlatQuant (Sun et al., 2024) mitigates online computational overhead by decomposing large matrices into Kronecker products of smaller matrices. This approach enables each transformer block to maintain its own transformation parameters, significantly enhancing the model's resilience to outliers following transformation and quantization. SVDQuant (Li et al., 2024b) employs a different strategy, extracting outliers from weights that absorb activation outliers through SVD decomposition and channeling these extracted outliers through low-rank LoRA branches. The method then quantizes the decomposed backbone weights while preserving the low-rank LoRA branches in full precision to minimize outlier impact on quantization performance. However, although optimized fused CUDA kernels have the potential to reduce the impact of additional overhead, certain modules (e.g., FFN1, FFN2) with extremely high intermediate dimensions still incur non-negligible computational costs. These costs arise from either persistently significant small matrix dimensions or relatively large low-rank parameter additions, substantially degrading single-operator performance in low-precision quantized settings and consequently impacting overall end-to-end inference latency.

To address these challenges, we propose SplitQuant, a novel approach that significantly reduces both the additional parameters and computational overhead by strategically splitting the in-channel dimension of linear layers and input activations while sharing a unified set of transformation parameters. Our method enables fine-grained control over the splitting granularity across different tensor shapes, thereby facilitating an effective trade-off between quantization error reduction and inference speed improvement. We make a critical observation that timestep modules in diffusion transformers amplify quantization errors during optimization, resulting in unstable convergence behavior. This phenomenon manifests consistently across image and video generation models and intensifies with increasing model depth. We propose an adaptive strategy to mitigate the amplification of quantization errors by timestep modules. Furthermore, to prevent gradient conflicts between text-guided and noise-based quantization losses during optimization in diffusion transformers, we introduce a grouped isolation optimization strategy that effectively improves quantization loss convergence. Through these innovations, we achieve state-of-the-art performance for low-bit diffusion models across text-to-image tasks. To ensure practical efficiency, we develop customized CUDA kernels optimized for various tensor shapes, delivering performance that closely matches standard W4A4

per-group quantization kernels while substantially accelerating the image generation pipeline. Our contributions are summarized as follows:

- **We introduce SplitQuant, a novel quantization approach that dramatically reduces computational overhead while maintaining model performance.** By strategically splitting the in-channel dimensions of linear layers and activations while sharing transformation parameters, SplitQuant achieves $2\times$ decrease in computational overhead compared to existing methods, rendering the additional costs virtually negligible.

- **We identify and address a critical yet previously overlooked challenge in diffusion transformer quantization.** Our analysis reveals that the inherent architectural properties of diffusion transformers fundamentally destabilize the quantization optimization process. We develop a specialized optimization pipeline that accounts for these unique structural characteristics, ensuring stable quantization loss convergence and significantly enhancing generation quality.

- **We demonstrate state-of-the-art results that establish new benchmarks for efficient diffusion model inference.** Through comprehensive evaluation on FLUX.1 and PixArt-$\Sigma$, we achieve truly lossless W4A4 quantization while delivering acceleration benefits that closely approach the theoretical limits of 4-bit quantization, making high-quality generative AI more accessible and practical.

## 2 RELATED WORK

**Diffusion Transformers.** Diffusion Transformers (Peebles & Xie, 2023) have emerged as the dominant architecture for visual generation, replacing convolutional U-Net backbones (Ronneberger et al., 2015) with transformer-based architectures. DiT pioneered this paradigm shift, demonstrating superior scalability through self-attention mechanisms. This architectural evolution has enabled significant advances across both image and video domains. In image synthesis, PixArt-$\alpha$ (Chen et al., 2023) and PixArt-$\Sigma$ (Chen et al., 2024) achieve high-quality text-to-image generation, while FLUX.1 (Labs, 2024) incorporates advanced architectural designs supporting diverse generation tasks. For video generation, Latte (Ma et al., 2024a) first adapts DiTs for temporal modeling, with subsequent models like Wan2.1 (Wan et al., 2025) scaling to unprecedented parameter counts while maintaining temporal coherence.

Recent work explores architectural convergence between language and vision models, enabling unified generation and understanding capabilities (Hurst et al., 2024; Comanici et al., 2025; Deng et al., 2025). This convergence facilitates advanced functionalities including image editing, long-context interaction, and multimodal generation. However, these capabilities incur substantial computational costs. Moreover, the iterative denoising process requires several forward passes, creating significant inference bottlenecks that motivate efficiency improvements through quantization.

**Quantization for Diffusion Models.** Model quantization addresses the computational demands of diffusion transformers by reducing numerical precision. Early diffusion quantization work focuses on temporal dynamics: Q-Diffusion (Li et al., 2023) and PTQ4DM (Shang et al., 2023) collect timestep-wise activation statistics to determine optimal quantization parameters. For transformer architectures, SmoothQuant (Xiao et al., 2023) redistributes activation outliers through channel-wise scaling, though performance degrades under ultra-low-bit settings (W4A4). Rotation-based methods (Quarot (Ashkboos et al., 2024), SpinQuant (Liu et al., 2024)) employ orthogonal transformations but fail in diffusion transformers due to timestep injection preventing matrix fusion (see Equation 1).

Recent DiT-specific approaches yield limited success. Q-DiT (Chen et al., 2025a) addresses channel imbalance via per-channel quantization, while PTQ4DiT (Wu et al., 2024b) designs fixed balance masks across timesteps. SVDQuant (Li et al., 2024b) isolates outliers through low-rank decomposition, and FlatQuant (Sun et al., 2024) reduces transformation overhead via Kronecker products. However, these methods incur substantial computational costs in high-dimensional layers, where transformation matrices or low-rank branches negate quantization benefits. Video generation presents additional challenges: ViDiT-Q (Zhao et al., 2024a) achieves lossless 8-bit quantization, while struggling with W4A4 settings.

Existing methods face two fundamental limitations. First, the interaction between timestep modules and the backbone creates optimization instabilities that current frameworks cannot address. Second,

outlier mitigation strategies introduce computational overhead that eliminates practical speedups, creating a gap between theoretical compression ratios and actual deployment performance. These challenges necessitate novel approaches that account for DiT's unique architectural properties while maintaining practical efficiency.

## 3 METHODOLOGY

In this paper, we first introduce the basic concepts of quantization and analyze the parameter and computational overhead of existing algorithms. We then propose SplitQuant, which reduces these overheads by partitioning the in-channel dimension of activations and weights. Finally, we develop a specialized optimization pipeline for diffusion transformer architectures that enhances training stability.

### 3.1 PRELIMINARY

Quantization reduces model size and computational overhead by mapping high-precision representations (fp32, fp16, bf16) to low-bit formats. Given tensor $X$, the process involves scaling to the target quantization range followed by rounding to a finite discrete set. Taking symmetric integer quantization as an example, this can be formalized as:

$$Q(X) = clamp\left(\left\lfloor \frac{X}{s} \right\rceil, -2^{n-1}, 2^{n-1} - 1\right), s = \frac{\max(|X|)}{2^{n-1} - 1}. \tag{2}$$

Where $clamp(\cdot)$ denotes the clipping operation, $s$ represent the quantization scale, $n$ is the target bit width, and $\lfloor \cdot \rceil$ denotes rounding. The quantized computation applies Equation 2 to both weights and activations, followed by dequantization after low-precision matrix multiplication. This process is formalized as:

$$Y = \hat{X}\hat{W} = s_x \cdot Q(X)Q(W) \cdot s_w. \tag{3}$$

Transformer-based inference suffers from outliers that become more severe with larger magnitudes and lower bit widths. While basic channel scaling methods (Xiao et al., 2023; Lin et al., 2024) show limited effectiveness in low-bit scenarios and rotation-based methods (Ashkboos et al., 2024; Liu et al., 2024) are incompatible with diffusion transformers (Section 1), two main approaches address this problem by minimizing additional parameter overhead.

SVDQuant (Li et al., 2024b) absorbs outliers into a low-rank branch, reducing quantization errors in the backbone network. Alternatively, nearest square root decomposition reduces rotation matrices to Kronecker products of smaller component matrices (Sun et al., 2024). However, for large in-channel dimensions, the decomposed matrices remain substantial. With $k$ factorized into $k_1 \times k_1$ and $k_2 \times k_2$ matrices, the overhead becomes $k_1^2 + k_2^2$ parameters and $2k_1k_2(m+n)(k_1+k_2)$ FLOPs. For $k = 24576, m = n = 4096$ ($k_1 = 128, k_2 = 192$), this introduces 128.8 GFLOPs per linear layer.

Given the compute-intensive nature of diffusion transformer inference, the aforementioned additional full-precision operations inevitably reduce the inference speed of quantized models. We address this by decomposing the in-channel dimension of weights and activations into slices that share common transformation parameters, thereby reducing both parameter and computational overhead.

### 3.2 SPLITQUANT

As illustrated in Figure 1, we propose SplitQuant, a novel quantization method applicable to all linear layers. The method operates by first partitioning the weight matrix $W$ and input tensor $X$ along the input channel dimension, yielding $s$ low-dimensional slices denoted as $W_i$ and $X_i$, where $i \in \{0, \cdots, s-1\}$. Each slice then undergoes independent fine-grained transformations to preserve information while reducing computational complexity. The transformed outputs from all slices are subsequently concatenated to produce the final result. Consequently, given the weight matrix $W$ and input tensor $X$, we formulate the optimization objective as follows:

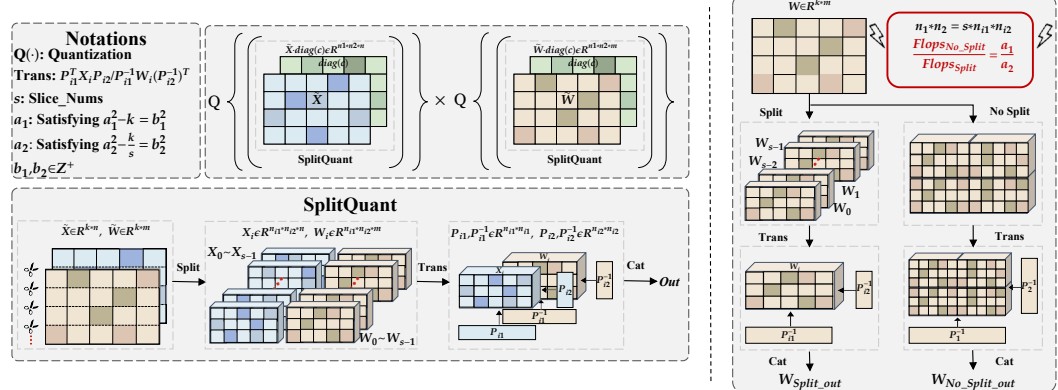

Figure 1: **Overview of SplitQuant. Left:** Detailed workflow of the proposed method. The weight matrix $W$ and input tensor $X$ are split along the input channel dimension into $s$ slices(The dots signify that intermediate slices are omitted). Each slice undergoes transformation using fine-grained projection matrices $P_{i1}$ and $P_{i2}$, and the resulting outputs are concatenated to produce the final result. **Right:** Comparative illustration of sliced versus non-sliced approaches. The slicing strategy achieves an $\frac{a_1}{a_2} \times = \frac{\sqrt{b_1^2 + k}}{\sqrt{b_2^2 + k}} \times \approx \sqrt{s} \times$ reduction in FLOPs compared to the non-sliced baseline.

$$\underset{P_0, \cdots, P_{s-1}}{\arg\min} \left\| XW^T - Q(\bigoplus_{i=0}^{s-1} X_i P_i) Q(\bigoplus_{i=0}^{s-1} P_i^{-1} W_i^T) \right\|_F^2. \tag{4}$$

where $\bigoplus$ denotes the concatenation operation across all $s$ slices, $P_i$ represents the transformation matrix for the $i$-th slice, and $||\cdot||_F$ denotes the Frobenius norm. To reduce computational complexity while maintaining expressiveness, each transformation matrix $P_i$ is further decomposed using the Kronecker product (Sun et al., 2024): $P_i = P_{i1} \otimes P_{i2}$. Consider input tensor $X \in \mathbb{R}^{n \times k}$ and weight matrix $W \in \mathbb{R}^{m \times k}$. The transformation matrix $P_i \in \mathbb{R}^{(k/s) \times (k/s)}$ is factorized into two smaller matrices $P_{i1} \in \mathbb{R}^{k_1 \times k_1}$ and $P_{i2} \in \mathbb{R}^{k_2 \times k_2}$, subject to the constraint $k_1 \times k_2 \times s = k$. The transformation of slices $W_i$ and $X_i$ using the Kronecker-factorized matrices $P_{i1}$ and $P_{i2}$ is performed as follows:

$$\mathcal{Q}(X_i P_i) \mathcal{Q}(P_i^{-1} W_i^T) = \mathcal{Q}(P_{i1}^T \hat{X}_i P_{i2}) \mathcal{Q}(P_{i1}^{-1} \hat{W}_i (P_{i2}^{-1})^T)^T. \tag{5}$$

where $\hat{X} \in \mathbb{R}^{n \times k_1 \times k_2}$ and $\hat{W} \in \mathbb{R}^{m \times k_1 \times k_2}$ denote the reshaped input and weight tensors, respectively. This formulation enables $\sqrt{s} \times$ reduction in the additional computational overhead introduced by the transformation matrices (see Appendix A.2 for detailed derivation).

To optimize the transformation parameters, we adopt the calibration strategy established in prior work Shao et al. (2023); Ma et al. (2024b); Sun et al. (2024). Specifically, we employ a small calibration dataset consisting of 128 image noise samples drawn uniformly at random across different diffusion timesteps. This calibration procedure independently optimizes the transformation matrices for each Transformer block to minimize reconstruction error. Formally, the optimization objective is defined as:

$$\Theta^* = \bigcup_{b \in \{\text{noise, text}\}} \underset{\Theta_b}{\arg\min} \left\| X_b^{\text{fp}} - X_b^{\text{quant}} \right\|_F^2 \tag{6}$$

Where $\Theta_b$ is the subset of learnable parameters corresponding to branch $b$, and each subset contains the corresponding transformation matrices $\{P_{i1}, P_{i2}, c, a_w, a_x\}$ for its respective branch. $X_b^{\text{fp}}$ and $X_b^{\text{quant}}$ denote the full-precision and quantized outputs of branch $b$. Specifically, $c$ denotes the per-channel scaling factors that mitigate the impact of outliers across different channels (Xiao et al.,

2023; Lin et al., 2024), while $a_w, a_x \in (0, 1)$ represent learnable clipping thresholds for weights and activations, respectively, enabling adaptive mapping to the target quantization range (Shao et al., 2023; Ma et al., 2024b; Sun et al., 2024). These complementary techniques are orthogonal to our primary approach and thus seamlessly integrated into our unified optimization framework.

However, a critical distinction arises when applying this framework to diffusion models. Unlike large language models that process sequential data unidirectionally, diffusion models incorporate text-guided conditioning branches and timestep modules, which introduce additional complexity to the optimization landscape. This architectural difference leads to instability in the convergence of the optimization process described above. To address this challenge, we introduce two stabilization strategies in the following section, specifically designed to ensure robust convergence in the presence of both multi-modal conditioning and timestep modules.

Figure 2: Activation MSE between the original (full-precision) model and the quantized model over all transformer blocks. The solid line denotes the MSE measured *before* the gating operation, whereas the dashed line denotes the MSE measured *after* the gate.

### 3.3 PipeLine

#### 3.3.1 Timestep Amplifies Quantization Errors

Diffusion models employ Adaptive Layer Normalization (AdaLN) modules (Perez et al., 2018; Peebles & Xie, 2023) to dynamically modulate feature representations based on temporal information. These modules integrate timestep embeddings and apply them to the model's latent features via element-wise affine transformations (multiplication and addition). Crucially, the AdaLN parameters exhibit substantial variation across both the temporal dimension (different timesteps) and the spatial dimension (different transformer blocks), resulting in diverse modulation patterns throughout the denoising process. This affine transformation mechanism exacerbates quantization-induced perturbations throughout the optimization process, thereby compromising the stability of loss convergence. Using the attention output features as an example, this mechanism can be formally expressed as:

$$\|X_{GM} \odot \mathcal{F}_{attn}(W, X) - X_{GM} \odot \mathcal{F}_{attn}(\mathcal{Q}(W), \mathcal{Q}(X), \Theta)\|_F^2. \tag{7}$$

Where $X_{GM}$ denotes the AdaLN module output $X_{gate\_msa}$, $\odot$ represents element-wise multiplication, and $\mathcal{F}_{attn}(\cdot)$ denotes the attention module within the transformer block. Equation 7 reveals that the mean squared quantization error of the attention output scales quadratically with the magnitude of the AdaLN modulation factors. This quadratic dependency has significant implications for quantization stability. As illustrated in Figure 2, AdaLN modules in deeper transformer blocks exhibit substantially larger absolute magnitudes, thereby severely amplifying quantization errors. These amplified errors then cascade through subsequent modules within each block, compounding the degradation of model performance.

Given that excessive AdaLN magnitudes can destabilize gradient-based optimization, we propose an adaptive learning rate strategy to enhance convergence stability. Observing that different diffusion transformer architectures exhibit distinct AdaLN magnitude distributions, we employ an architecture-aware tuning strategy. Specifically, we assign lower learning rates to models with larger average AdaLN magnitudes to mitigate gradient instability and ensure robust convergence.

#### 3.3.2 Text-Guided Interference with Optimization Direction

In diffusion models, text conditioning guides controllable content generation, creating a dual-loss optimization landscape: noise reconstruction loss and text alignment loss. Parameters preceding the attention module receive gradients from both objectives simultaneously through the attention's cross-modual computation. As illustrated in Figure 3(left), this gradient interference causes parameters to diverge from either optimal point, degrading quantization performance. To address this

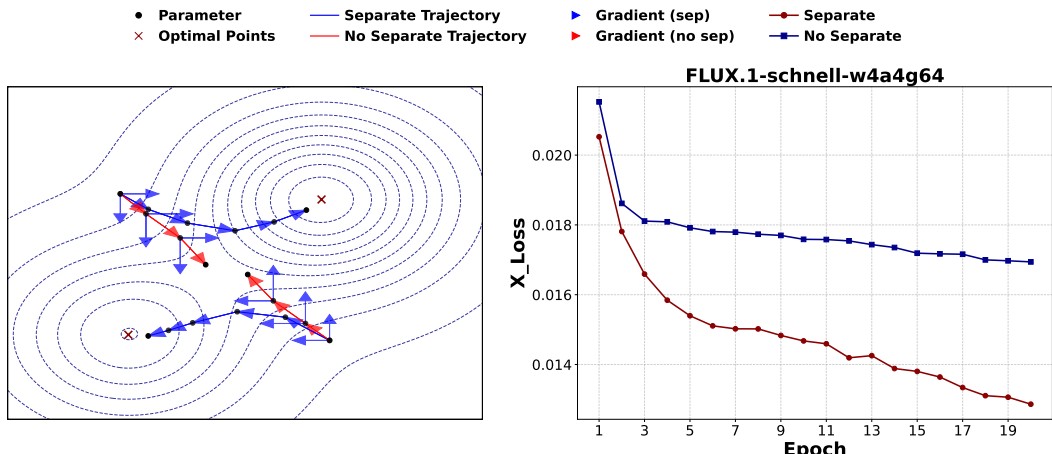

Figure 3: Illustration of separation behavior in optimization. **Left:** Loss landscape with two minima. Blue trajectories separate and converge to individual optima; red ones fail to separate. Arrows show gradient directions. **Right:** Training curves of $X\_Loss$ over 20 epochs in block 8 of FLUX.1-schnell under w4a4g64 quantization. "Separate" converges faster and achieves lower loss than "No Separate".

conflict, we introduce a grouped optimization strategy that decouples the gradient flow: main branch parameters optimize exclusively against the noise loss, while text branch parameters target only the text-conditioning loss. This gradient isolation achieves superior loss convergence and quantization accuracy, as demonstrated in Figure 3(right).

## 4 EXPERIMENTS

### 4.1 IMPLEMENTATION DETAILS

**Models.** We evaluate a set of state-of-the-art generative diffusion models built upon the Diffusion Transformer (DiT) architecture, spanning both text-to-image and text-to-video tasks.

- **FLUX.1**: As the state-of-the-art open-source diffusion model, it employs a substantial architecture of 19 dual-stream blocks and 38 single-stream blocks, totaling 12B parameters. For our evaluation, we selected its 4-step time-step distilled variant—FLUX.1-schnell (Labs, 2024).
- **PixArt-$\Sigma$**: Another advanced diffusion model based on the DiT architecture, features a stack of 28 attention blocks, each combining self-attention, cross-attention, and feed-forward layers, contains approximately 600M parameters. We evaluate PixArt-$\Sigma$ (Chen et al., 2024) using its default 20-step inference setting.
- **Wan2.1**: A state-of-the-art text-to-video diffusion model based on the DiT architecture, composed of a stack of 40 attention blocks and with approximately 14B parameters. We evaluate Wan2.1 (Wan et al., 2025) under its default 50-step inference configuration.

**Datasets.** Following Prior work (Li et al., 2023; Zhao et al., 2024b), we randomly sample 128 prompts from the COCO Captions 2024 dataset (Lin et al., 2014) for model calibration. To evaluate generalization, we select 5K prompts each from the MJHQ-30K (Li et al., 2024a) and sDCI datasets (Urbanek et al., 2024) for text-to-image (T2I) benchmarking.

- **MJHQ-30K** : This dataset consists of 10 common categories, with 3,000 diverse samples per category, enabling a robust evaluation of model aesthetics.
- **sDCI** : The original DCI dataset comprises 7,805 images, each accompanied by extensive human annotations (averaging over 1,000 words). To meet the 77-token input limit of diffusion models, a condensed version, sDCI, was created to assess the ability of models to generate realistic images from semantically rich yet concise descriptions.

**Baselines.** We compared SplitQuant with SVDQuant. SVDQuant (Li et al., 2024b) is an efficient post-training quantization method that effectively absorbs outliers in weights and activations through low-rank decomposition, thereby significantly reducing memory usage while maintaining image quality.

**Metrics.** To comprehensively evaluate our model's performance on the text-to-image (T2I) task, we adopt multiple quantitative metrics. Specifically, we use Fréchet Inception Distance (FID) (Heusel et al., 2017) and Image Reward (IR) (Xu et al., 2023) to measure the distributional divergence between generated and real images and to approximate human preferences. Additionally, Learned Perceptual Image Patch Similarity (LPIPS) is employed to assess perceptual consistency (Zhang et al., 2018), while Peak Signal-to-Noise Ratio (PSNR) quantifies pixel-level fidelity and the preservation of brightness information.For text-to-video (T2V) evaluation, we adopt VBench (Huang et al., 2023)——a comprehensive benchmark that evaluates video generation quality across multiple dimensions including motion fidelity, temporal consistency, and text-video alignment—and report its per-dimension scores.

**Setting Details.** During training, we uniformly set the number of epochs to 20 for all models. The learning rates were set to 0.01 for FLUX.1-schnell, 0.007 for PixArt-$\Sigma$, and 0.0008 for Wan2.1. During quantization, both activations and weights are symmetrically quantized per group, with a group size of 64 and 16-bit scale; the low-rank branch uses a rank size of 32. Except for the FLUX.1 model, the key and value projections in cross-attention are preserved in 16-bit precision. These configurations remain consistent with those of SVDQuant.

## 4.2 RESULTS

**comparative experiment.** Table 1 presents the quantitative evaluation results of different quantization methods and settings on the text-to-image (T2I) task. The data clearly shows that our 4-bit quantization method outperforms SVDQuant in almost all metrics. Under specific configurations, our method achieves near-lossless quantization.

Table 1: Quantitative comparison of image generation quality and similarity metrics across different models and quantization methods on MJHQ and sDCI datasets.Here, (GPTQ) and (RTN) denote the quantization schemes employed during inference.

| Model | Precision | Method | MJHQ | | | | sDCI | | | |
| | | | Quality | | Similarity | | Quality | | Similarity | |
| | | | FID ($\downarrow$) | IR ($\uparrow$) | LPIPS ($\downarrow$) | PSNR ($\uparrow$) | FID ($\downarrow$) | IR ($\uparrow$) | LPIPS ($\downarrow$) | PSNR ($\uparrow$) |
|---|---|---|---|---|---|---|---|---|---|---|
| **FLUX.1 -schnell** (4 Steps) | **BF16** | – | 18.9 | 0.986 | – | – | 20.6 | 0.987 | – | – |
| | **w4a4** | **SVDQuant** | **18.2** | 0.972 | 0.258 | 18.5 | **20.0** | 0.994 | 0.262 | 17.3 |
| | | **Ours (GPTQ)** | 18.7 | **0.981** | **0.241** | **18.8** | 20.6 | 0.994 | **0.243** | **17.8** |
| | | **Ours (RTN)** | 18.7 | **0.981** | 0.260 | 18.3 | 20.4 | **0.999** | 0.266 | 17.2 |
| **PixArt-$\Sigma$** (20 Steps) | **FP16** | – | 16.1 | 0.990 | – | – | 24.5 | 0.987 | – | – |
| | **w4a4** | **SVDQuant** | 19.6 | 0.886 | **0.360** | **16.8** | 28.1 | 0.896 | **0.401** | 15.4 |
| | | **ViDiT-Q** | 412 | -2.27 | 0.854 | 6.44 | 425 | -2.28 | 0.838 | 6.70 |
| | | **PTQ4DiT** | 309 | -2.27 | 0.938 | 9.75 | 332 | -2.28 | 0.895 | 9.44 |
| | | **FlatQuant** | 21.7 | 0.917 | 0.390 | 16.4 | 25.8 | 0.952 | 0.420 | 15.3 |
| | | **Ours (GPTQ)** | 20.5 | **0.950** | 0.378 | 16.7 | 25.1 | 0.971 | 0.403 | **15.7** |
| | | **Ours (RTN)** | 21.0 | 0.956 | 0.402 | 16.2 | **24.9** | **0.979** | 0.431 | 15.0 |

On the FLUX.1-schnell model, SplitQuant consistently outperforms SVDQuant in both LPIPS and PSNR, indicating superior visual fidelity and structural similarity in the generated images. For the PixArt-$\Sigma$ model, SplitQuant consistently exceeds SVDQuant on the IR (Image Reward) metric, demonstrating that its outputs better align with human aesthetic preferences.

Table 2 presents the quantitative evaluation on the Wan2.1 video generation model using VBench metrics. SplitQuant maintains competitive performance in text-to-video tasks, achieving an average score of 69.29, which is comparable to the BF16 baseline (70.11) and slightly superior to SVDQuant (69.08). proving its effectiveness in preserving temporal coherence and visual quality in complex video generation tasks.

Table 2: Comparison with SVDQuant on the VBench benchmark.We selected 8 out of the 16 evaluation dimensions from VBench and report the average score. Higher metrics indicate better performance. The results for "Ours" are obtained with a slice number of 4.

| Method | Precision | Imaging Quality | Aesthetic Quality | Motion Smooth | Dynamic Degree | BG. Consist | Subject. Consist | Scene Consist | Overall Consist | Avg. |
|---|---|---|---|---|---|---|---|---|---|---|
| – | **BF16** | 66.17 | 62.71 | 97.30 | 94.44 | 95.85 | 92.78 | 26.16 | 25.47 | 70.11 |
| **SVDQuant** | **W4A4** | 65.66 | 61.88 | 96.93 | 84.72 | 95.49 | 91.91 | **30.52** | 25.49 | 69.08 |
| **Ours (GPTQ)** | **W4A4** | **66.17** | **62.23** | 97.32 | 81.94 | 95.42 | **93.68** | 26.74 | 25.19 | 68.59 |
| **Ours (RTN)** | **W4A4** | 64.21 | 61.60 | **97.36** | **88.89** | **95.72** | 92.67 | 28.20 | **25.68** | **69.29** |

To further verify the broad applicability of SplitQuant, we also evaluated it on vision-language models and large language models; details are provided in the appendix A.3

**Ablation study.** Table 3 presents the evaluation results of our method under varying numbers of slices (SliceNums) and different quantization error optimization strategies. The results show that as the number of slices increases, all evaluation metrics remain largely stable—sometimes even exhibiting slight improvements—indicating nearly lossless model performance, with trends visualized in Figure 4(More details can be found in Appendix A.4). Furthermore, we systematically analyze the impact of different quantization error optimization strategies on model performance. In the Flux.1 model, the "linear-wise" strategy effectively avoids interference with gate_msa and gate_mlp, leading to improved generation quality. In contrast, the "tanh" method, which applies a tanh mapping to the outputs of gate_msa and gate_mlp, shifts the optimization trajectory and consequently degrades performance. In the PixArt-$\Sigma$ model, however, quantization error spikes are less pronounced, resulting in relatively minor performance differences across optimization strategies. In summary, SplitQuant achieves an excellent trade-off between computational efficiency and generation fidelity, maintaining high-quality image synthesis even under substantially reduced computational complexity.

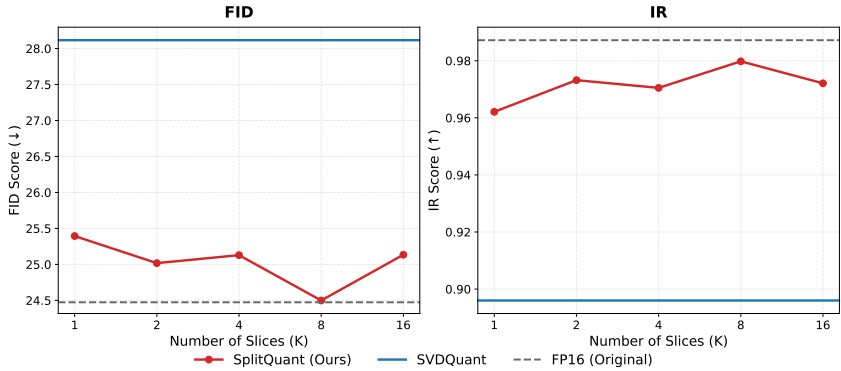

Figure 4: FID and Image Reward (IR) scores on the sDCI dataset for PixArt-$\Sigma$ under different numbers of slices (K).

Table 3: Quantitative evaluation of image generation quality and similarity for different models on the MJHQ and sDCI datasets, comparing the impact of varying slice numbers (SliceNums) and optimization strategies on model performance. Specifically, "linear-wise" denotes independent calibration of each Linear layer, while "tanh" indicates applying a tanh mapping to the outputs of gate_msa and gate_mlp.

| model | precision | method | SliceNums | MJHQ | | | | sDCI | | | |
| | | | | Quality | | Similarity | | Quality | | Similarity | |
| | | | | FID ($\downarrow$) | IR ($\uparrow$) | LPIPS ($\downarrow$) | PSNR ($\uparrow$) | FID ($\downarrow$) | IR ($\uparrow$) | LPIPS ($\downarrow$) | PSNR ($\uparrow$) |
|---|---|---|---|---|---|---|---|---|---|---|---|
| FLUX.1 -schnell | w4a4 | Ours (GPTQ) | 1 | 18.6 | 0.981 | **0.239** | 18.9 | 20.5 | 0.993 | 0.243 | **17.8** |
| | | | 2 | 18.5 | 0.983 | 0.241 | 18.8 | 20.4 | **1.000** | **0.242** | **17.8** |
| | | | 4 | 18.7 | 0.981 | 0.241 | 18.8 | 20.6 | 0.994 | 0.243 | **17.8** |
| | | Ours (RTN) | 1 | 18.6 | **0.991** | 0.261 | 18.3 | 20.7 | 0.996 | 0.266 | 17.2 |
| | | | 2 | 18.6 | 0.989 | 0.261 | 18.4 | **20.1** | **1.000** | 0.265 | 17.3 |
| | | | 4 | 18.7 | 0.981 | 0.260 | 18.3 | 20.4 | 0.999 | 0.265 | 17.2 |
| | | linear-wise | – | **18.3** | 0.982 | **0.239** | 18.9 | 20.3 | 0.993 | 0.244 | 17.7 |
| | | tanh | – | 22.1 | 0.794 | 0.522 | 13.3 | 24.6 | 0.899 | 0.549 | 12.7 |
| PixArt-$\Sigma$ | w4a4 | Ours (GPTQ) | 1 | 20.7 | 0.950 | 0.371 | 17.0 | 25.4 | 0.962 | 0.400 | **15.8** |
| | | | 2 | **20.5** | 0.949 | 0.372 | 16.9 | 25.0 | 0.973 | 0.398 | **15.8** |
| | | | 4 | **20.5** | 0.950 | 0.378 | 16.7 | 25.1 | 0.971 | 0.403 | 15.7 |
| | | Ours (RTN) | 1 | 20.9 | 0.943 | 0.389 | 16.3 | **24.8** | 0.967 | 0.417 | 15.1 |
| | | | 2 | 21.6 | 0.914 | 0.387 | 16.5 | 26.0 | 0.943 | 0.420 | 15.4 |
| | | | 4 | 21.0 | **0.956** | 0.402 | 16.2 | 24.9 | **0.979** | 0.431 | 15.0 |
| | | linear-wise | – | 23.3 | 0.922 | 0.390 | 16.9 | 27.3 | 0.961 | 0.427 | 15.6 |
| | | tanh | – | 20.6 | 0.953 | **0.360** | **17.2** | 25.3 | 0.978 | **0.391** | **15.8** |

## 5 CONCLUSION

This work presents SplitQuant, an efficient low-bit quantization method for Diffusion Transformers (DiTs) that slices weights and activations along the input channel dimension while sharing transformation parameters across slices. This approach reduces computational overhead by approximately 50% compared to existing methods while effectively mitigating outlier-induced quantization errors. We identify and address two critical challenges in diffusion model quantization: (1) error amplification by timestep modules causing optimization instability, and (2) gradient conflicts between text-guided objectives and noise reconstruction loss. Our solution employs adaptive learning rate scheduling to counteract timestep-induced error amplification and grouped isolation optimization to decouple conflicting multi-objective gradients. Extensive experiments demonstrate that SplitQuant achieves near-lossless W4A4 quantization on state-of-the-art DiT models including FLUX.1, PixArt-$\Sigma$, and Wan2.1. Through custom-optimized CUDA kernels, we minimize the overhead of additional parameters and quantization operations, enabling end-to-end inference acceleration approaching the theoretical 4-bit limit. SplitQuant thus provides a practical, high-performance solution for deploying high-resolution diffusion models efficiently.

## 6 ETHICS STATEMENT

This work complies with the ICLR Code of Ethics. The study does not involve human subjects or animal experimentation. All datasets used were obtained and utilized in accordance with their respective usage policies, ensuring full respect for privacy rights. We have taken deliberate steps to mitigate potential biases and avoid discriminatory outcomes throughout the research process. No personally identifiable information was employed, and no experiments were conducted that could pose privacy or security risks. We uphold the principles of transparency and research integrity at every stage of this work.

## 7 REPRODUCIBILITY STATEMENT

We have made every effort to ensure the reproducibility of the results reported in this paper. To support replication and verification, all code and datasets have been made publicly available in an anonymous repository. The paper provides a comprehensive description of the experimental setup. We believe these measures will empower other researchers to reproduce our findings and contribute to the continued advancement of the field.

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

# A  APPENDIX

## A.1  LLM USAGE STATEMENT

Large Language Models (LLMs) were used to aid in the writing and polishing of the manuscript. Specifically, we used an LLM to assist in refining the language, improving readability, and ensuring clarity in various sections of the paper. The model helped with tasks such as sentence rephrasing, grammar checking, and enhancing the overall flow of the text.

It is important to note that the LLM was not involved in the ideation, research methodology, or experimental design. All research concepts, ideas, and analyses were developed and conducted by the authors. The contributions of the LLM were solely focused on improving the linguistic quality of the paper, with no involvement in the scientific content or data analysis.

The authors take full responsibility for the content of the manuscript, including any text generated or polished by the LLM. We have ensured that the LLM-generated text adheres to ethical guidelines and does not contribute to plagiarism or scientific misconduct.

## A.2  DERIVATION OF THE OVERHEAD COMPRESSION RATIO

To analyze the computational overhead of transformation matrices in SplitQuant, we first clarify the notation:

- $K$: number of slices along the in-channel dimension;
- $b$: batch size;
- $s$: sequence length;
- $n_1, n_2$: dimensions of the Kronecker factors in the *non-split* baseline, satisfying $n_1 n_2 = k$, where $k$ is the original in-channel dimension;
- $n_{i1}, n_{i2}$: dimensions of the Kronecker factors for the $i$-th slice in SplitQuant, satisfying $n_{i1} n_{i2} = k/K$ for all $i$;
- All slices are of equal size, so $n_{i1} = n_{j1}$ and $n_{i2} = n_{j2}$ for any $i, j$;
- Auxiliary scalars $a_1, a_2$ are defined to capture the effective scale of transformations:

$$a_1^2 - k = b_1^2, \quad a_2^2 - \frac{k}{K} = b_2^2, \quad \text{with } b_1, b_2 \in \mathbb{Z}^+.$$

  Under symmetric factorization ($n_1 = n_2 = \sqrt{k}$, $n_{i1} = n_{i2} = \sqrt{k/K}$), we have $a_1 \propto \sqrt{k}$ and $a_2 \propto \sqrt{k/K}$, implying $a_1/a_2 \approx \sqrt{K}$.

The FLOPs introduced by the transformation operations are then given by:

$$F_{\text{Split}} = \sum_{i=1}^{K} 2\, b\, s\, n_{i1} n_{i2}(n_{i1} + n_{i2}), \tag{8}$$

$$F_{\text{No\_Split}} = 2\,b\,s\,n_1 n_2(n_1 + n_2). \tag{9}$$

Therefore, the speedup ratio is:

$$\frac{F_{\text{No\_Split}}}{F_{\text{Split}}} = \frac{2\,b\,s\,n_1 n_2(n_1 + n_2)}{\sum\limits_{i=1}^{K} 2\,b\,s\,n_{i1} n_{i2}(n_{i1} + n_{i2})} \tag{10}$$

$$= \frac{n_1 n_2(n_1 + n_2)}{\sum\limits_{i=1}^{K} n_{i1} n_{i2}(n_{i1} + n_{i2})} \tag{11}$$

$$= \frac{K \cdot (n_1 + n_2)}{K \cdot (n_{i1} + n_{i2})} \quad (\text{since } n_1 n_2 = k n_{i1} n_{i2}) \tag{12}$$

$$= \frac{n_1 + n_2}{n_{i1} + n_{i2}} \tag{13}$$

$$= \frac{a_1}{a_2} \tag{14}$$

$$\approx \sqrt{K}. \tag{15}$$

## A.3  EXTENDED EXPERIMENTS ON LANGUAGE MODELS

We conducted experiments on vision-language models (VLMs) and large language models (LLMs) to verify the broad applicability of SplitQuant.As shown in Table 4, SplitQuant consistently outperforms FlatQuant on both Qwen-VL-Chat and MiniCPM-V across all metrics, approaching or even surpassing the BF16 baseline in most cases.

Table 4: Evaluation on vision-language models under W4A4 + KV4 quantization.

| Model | Method | MME-P | MME-C | MMBench | MMStar | TextVQA | SEEDBench | PPL↓ |
|---|---|---|---|---|---|---|---|---|
| | BF16 | 1429 | 392 | 56.1 | 33.5 | 60.6 | 63.63 | 9.85 |
| Qwen-VL | FlatQuant | 1420 | 424 | 55.1 | 35.3 | 57.7 | 62.15 | 10.88 |
| | **SplitQuant** | **1459** | **424** | **57.1** | **36.9** | **59.7** | **62.23** | **10.74** |
| | BF16 | 1421 | 304 | 64.2 | 37.9 | 61.7 | 65.2 | 8.11 |
| MiniCPM-V | FlatQuant | 1228 | 308 | 44.8 | 35.3 | 52.1 | 54.5 | 8.61 |
| | **SplitQuant** | **1337** | **332** | **58.7** | **38.1** | **60.5** | **63.2** | **8.48** |

Table 5 shows that SplitQuant achieves the highest average accuracy (70.60%) and the lowest perplexity (7.83) on Qwen2.5-7B-Instruct, and outperforms BF16 on several reasoning benchmarks such as ARC-E and Winogrande.

Table 5: Language modeling performance on Qwen2.5-7B-Instruct under W4A4 + KV4 quantization. Accuracy (%) and perplexity (PPL) are reported. Higher accuracy and lower PPL are better.

| Method | ARC-C | ARC-E | HellaSwag | LAMBADA | Winogrande | Avg Acc ↑ | PPL ↓ |
|---|---|---|---|---|---|---|---|
| BF16 | 51.62% | 75.88% | 79.70% | 67.81% | 68.75% | 68.75% | 8.35 |
| FlatQuant | 54.52% | 77.48% | 78.02% | 64.62% | 67.72% | 68.47% | 8.18 |
| **SplitQuant** | **55.38%** | **80.93%** | **79.24%** | **68.08%** | **69.38%** | **70.60%** | **7.83** |

## A.4  ABLATION STUDY ON SLICE NUMBER AND QUANTIZATION ERROR BEHAVIOR

To investigate how the number of slices (SliceNums) affects performance, we conducted an ablation study on the PixArt-Σ model using the sDCI dataset. Results are summarized in Table 6.

Table 6: Quantitative results of SplitQuant under different slice numbers on the sDCI dataset (PixArt-$\Sigma$, w4a4 quantization).

| Method | SliceNums | FID↓ | IR↑ | LPIPS↓ | PSNR↑ |
|---|---|---|---|---|---|
| FP16 | – | 24.4733 | 0.9872 | – | – |
| SVDQuant | – | 28.1159 | 0.8960 | 0.4007 | 15.4369 |
| Ours (GPTQ) | 1 | 25.3941 | 0.9621 | 0.4001 | **15.7972** |
| Ours (GPTQ) | 2 | 25.0178 | 0.9732 | **0.3976** | 15.7502 |
| Ours (GPTQ) | 4 | 25.1289 | 0.9705 | 0.4030 | 15.6865 |
| Ours (GPTQ) | 8 | **24.4986** | **0.9798** | 0.3992 | 15.5903 |
| Ours (GPTQ) | 16 | 25.1344 | 0.9721 | 0.4231 | 15.3846 |

It can be observed that as the number of slices increases, model performance degrades only gradually—for instance, the FID score increases only slightly from 25.0178 with 2 slices to 25.1344 with 16 slices, indicating that quantization error accumulates very slowly. In practice, using 2 to 4 slices strikes an optimal balance between quantization error suppression and computational overhead. More importantly, even with as many as 16 slices, our method still maintains strong performance (SplitQuant achieves an FID of 25.1344, significantly better than SVDQuant's 28.1159), demonstrating remarkable robustness. From a theoretical perspective, although the transformation capacity of each individual slice is reduced, the increased number of slices provides a greater number of independent transformation degrees of freedom, offering compensatory representational flexibility. Consequently, each slice can still learn meaningful, outlier-aware quantization transformations, allowing the performance benefits of quantization to persist even as the number of slices grows—without rapidly diminishing.

## A.5 QUALITATIVE RESULTS

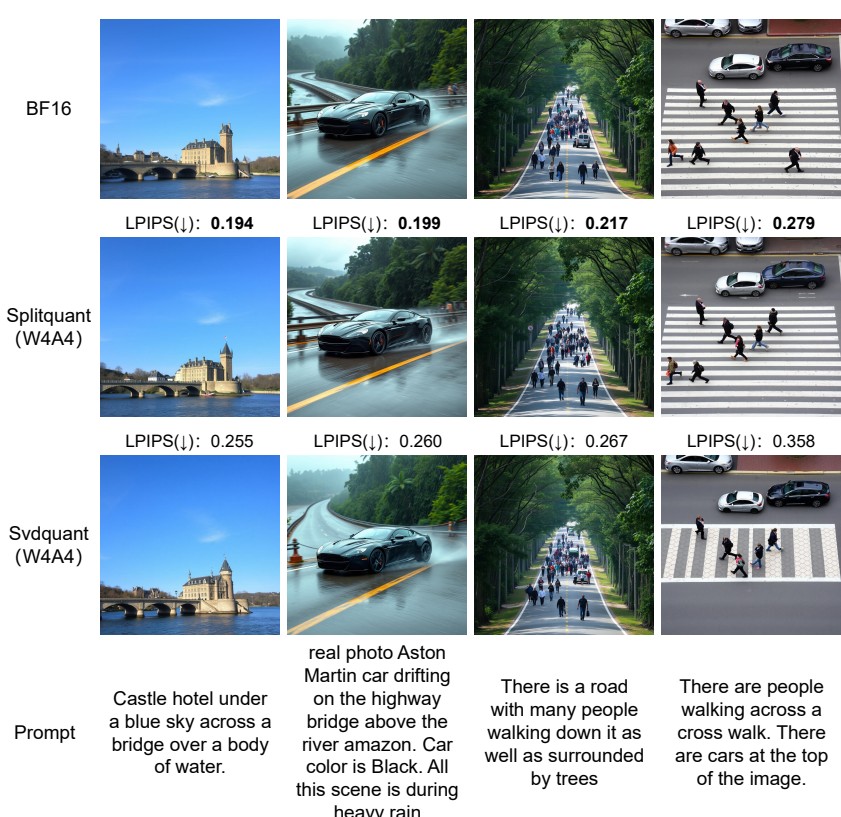

Figure 5: Qualitative comparison on FLUX.1-schnell. Visual comparisons of text-to-image generation results under W4A4 quantization. SplitQuant preserves fine-grained details.

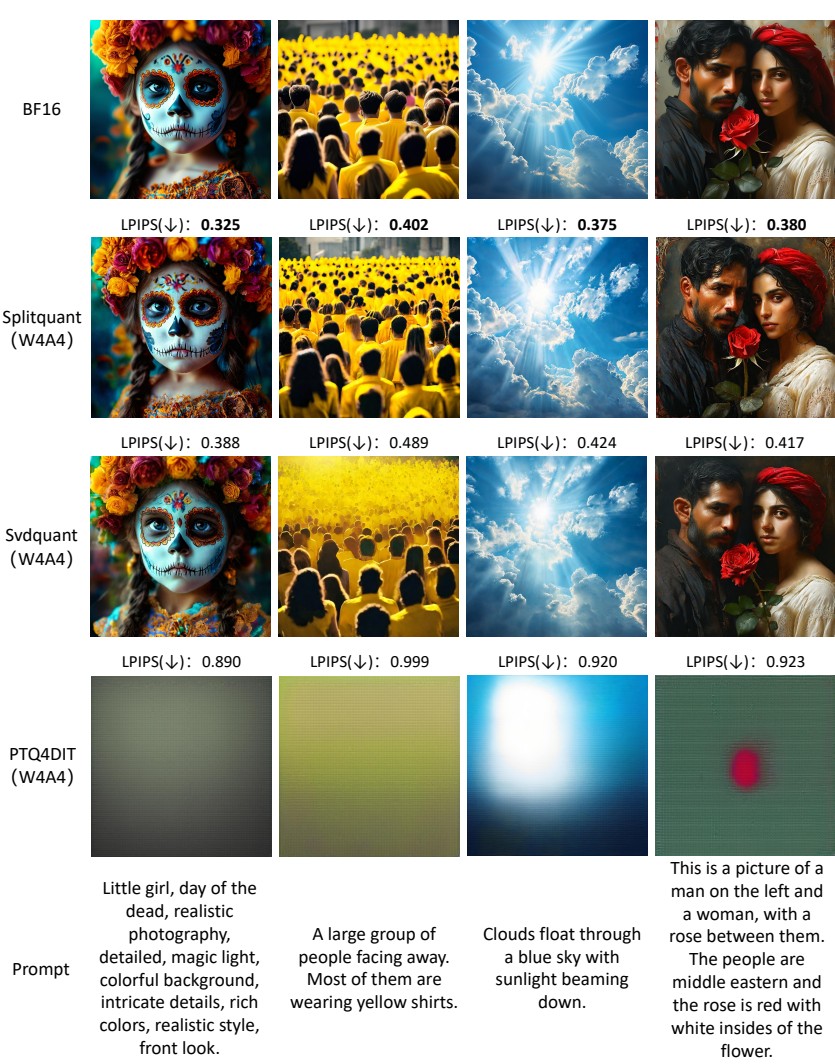

Figure 6: Qualitative comparison on PixArt-Σ. Comparison of generation quality across different quantization methods. While PTQ4DiT suffers from severe quantization collapse, generating unintelligible noise images, SplitQuant maintains high visual quality comparable to the full-precision model.

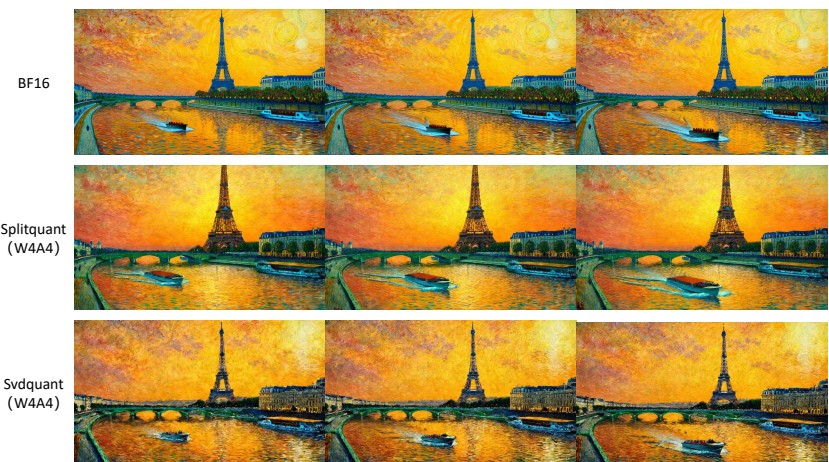

Figure 7: Qualitative comparison on Wan2.1 (Clarity). SplitQuant generates frames with sharper edges and clearer structures, avoiding the blurriness observed in SVDQuant results.

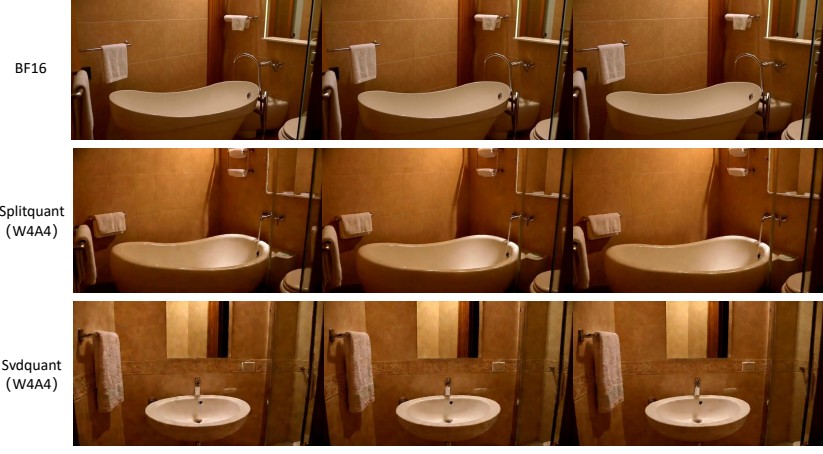

Figure 8: Qualitative comparison on Wan2.1 (Semantic Accuracy). SplitQuant accurately reflects the prompt's semantic requirements, whereas SVDQuant exhibits noticeable deviations from the text description.

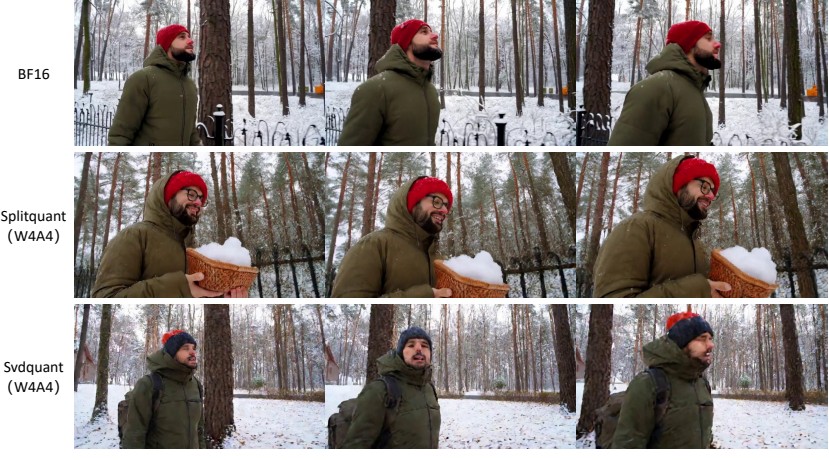

Figure 9: Qualitative comparison on Wan2.1 (Facial Fidelity).SplitQuant effectively reconstructs facial features with high fidelity, significantly outperforming SVDQuant, which suffers from severe degradation in facial regions.

