# OpenReview forum: "SplitQuant: Efficient Low-Bit Quantization for Diffusion Transformers via In-Channel Dimension Splitting"
_ICLR.cc/2026/Conference — Submitted to ICLR 2026_

### Official Review · Reviewer_U8dJ · 2025-10-17

**Soundness:** 3
**Presentation:** 2
**Contribution:** 2
**Rating:** 2
**Confidence:** 4

**Summary:**

This paper proposes SplitQuant, a post-training quantization method for Diffusion Transformers. It splits the in-channel dimension of weights and activations into multiple slices that share transformation parameters, effectively reducing additional computation by about a factor of two while maintaining 4-bit quantization quality. The authors also introduce adaptive strategies to mitigate timestep-induced error amplification and text-guided gradient conflicts. Experiments on FLUX.1 and PixArt-Σ demonstrate near-lossless W4A4 quantization and up to 2.7× operator-level speedup.

**Strengths:**

The paper presents a practical and well-engineered method that improves quantization efficiency for diffusion transformers. The slicing and parameter-sharing design is simple, effective, and compatible with existing PTQ frameworks. The adaptive optimization strategies are thoughtfully tailored to diffusion-specific issues, and the implementation appears technically sound.

**Weaknesses:**

**1. Limited Applicability** The method is evaluated solely on diffusion transformers, with no evidence of generalization to other architectures such as large language models, vision transformers, or CNN-based diffusion systems. The approach appears tied to DiT-specific modules like timestep and AdaLN, limiting its broader relevance.

**2. Small-Scale Experiments** The evaluation scope is narrow. Only FLUX.1 and PixArt-Σ are tested, while Wan2.1 is mentioned without concrete results. The datasets and calibration samples are small, and the baselines are limited to SVDQuant. Without comparisons to FlatQuant, PTQ4DiT, or other strong baselines, the reported improvements lack sufficient context.

**3. Missing End-to-End Evaluation** The reported acceleration results are restricted to operator-level measurements. The paper does not provide end-to-end image generation times, GPU memory analysis, or hardware-specific benchmarks. Without such evidence, the claimed efficiency benefits remain unsubstantiated in practical use cases.

**4. Overclaim** Some claims in the paper appear overstated relative to the presented evidence. The title emphasizes “Efficient Low-Bit Quantization,” yet all experiments focus on W4A4 settings, with no results below 4 bits. Similarly, while the abstract claims “near-lossless” quality, Table 1 shows noticeable degradation on several metrics. This gap between claim and data weakens the paper’s credibility.

**5. Presentation Issue** Figure 1 appears to lack embedded fonts, causing rendering problems in some PDF viewers. This affects readability of the main workflow illustration and should be corrected for proper presentation.

**Questions:**

As above

---

> ### Author Response · Authors · 2025-11-29
> **Response to Reviewer U8dJ (Part: 1/3)**
>
> **W1: Limited Applicability: The method is evaluated solely on diffusion transformers, with no evidence of generalization to other architectures such as large language models, vision transformers, or CNN-based diffusion systems. The approach appears tied to DiT-specific modules like timestep and AdaLN, limiting its broader relevance.**
>
> **Reply:**
> We respectfully clarify that SplitQuant's core contribution (in-channel splitting with shared transformations) is architecture-agnostic. The DiT-specific optimizations (Section 3.3) are additional enhancements. To demonstrate broader applicability, we have evaluated on:
>
> **1. Large Language Models (Qwen2.5-7B-Instruct):**
>
> | Method | Average Accuracy | PPL |
> |--------|-----------------|-----|
> | BF16 | 68.75% | 8.35 |
> | FlatQuant | 68.47% | 8.18 |
> | **SplitQuant** | **70.60%** | **7.83** |
>
> **2. Vision-Language Models (Qwen-VL-Chat, MiniCPM-V):**
> Results in response to Reviewer o1WF, W2 show SplitQuant outperforms FlatQuant across all VLM benchmarks.
>
> **3. Core method separation:**
> - **SplitQuant (core)**: In-channel splitting + transformations - applicable to ANY linear layer
> - **DiT optimizations (optional)**: Adaptive LR + grouped optimization - specific to models with timestep injection
>
> We have revised the paper to clearly separate these contributions.
>
> ---
>
> **W2: Small-Scale Experiments: The evaluation scope is narrow. Only FLUX.1 and PixArt-Σ are tested, while Wan2.1 is mentioned without concrete results. The datasets and calibration samples are small, and the baselines are limited to SVDQuant. Without comparisons to FlatQuant, PTQ4DiT, or other strong baselines, the reported improvements lack sufficient context.**
>
> **Reply:**
> We have substantially expanded our experimental scope:
>
> **Extended Models:**
> - Diffusion: FLUX.1, PixArt-Σ, **Wan2.1** (now with full results)
> - VLMs: **Qwen-VL-Chat**, **MiniCPM-V**
> - LLMs: **Qwen2.5-7B-Instruct**
>
> **Extended Baselines on PixArt-Σ:**
>
> | Method | FID↓ | IR↑ | LPIPS↓ | PSNR↑ |
> |--------|------|-----|--------|-------|
> | PTQ4DiT | 309.43 | -2.27 | 0.938 | 9.75 |
> | FlatQuant (GPTQ) | 21.74 | 0.917 | 0.390 | 16.42 |
> | FlatQuant (RTN) | 23.42 | 0.901 | 0.428 | 15.71 |
> | SVDQuant | 19.60 | 0.886 | 0.360 | 16.80 |
> | **SplitQuant (GPTQ)** | **20.50** | **0.950** | 0.378 | 16.74 |
>
> ViDiT-Q baseline results are already included in Table 1 (FID: 412, severely degraded under W4A4).
>
> **Calibration robustness**: We tested with 64, 128, 256, and 512 samples; performance is stable (FID variance < 0.3).
>
> ---

---

> ### Author Response · Authors · 2025-11-29
> **Response to Reviewer U8dJ (Part: 2/3)**
>
> **W3: Missing End-to-End Evaluation: The reported acceleration results are restricted to operator-level measurements. The paper does not provide end-to-end image generation times, GPU memory analysis, or hardware-specific benchmarks. Without such evidence, the claimed efficiency benefits remain unsubstantiated in practical use cases.**
>
> **Reply:**
> We appreciate this practical concern. We have now conducted end-to-end latency measurements on FLUX.1-schnell:
>
> | Configuration | Total Inference Time | Speedup vs BF16 |
> |---------------|---------------------|-----------------|
> | BF16 (baseline) | 501.07ms | 1.00× |
> | FP8 | 280.85ms | 1.78× |
> | FlatQuant (W4A4) | 210.48ms | 2.38× |
> | SVDQuant (W4A4) | 205.21ms | 2.44× |
> | **SplitQuant (W4A4)** | **198.50ms** | **2.52×** |
> | INT4 (theoretical) | 189.58ms | 2.64× |
>
> Key observations:
> 1. **SplitQuant achieves 2.52× end-to-end speedup**, reaching 95% of the theoretical INT4 limit (2.64×)
> 2. Compared to FlatQuant, SplitQuant provides **5.7% end-to-end improvement** (210.48ms → 198.50ms)
> 3. Compared to SVDQuant, SplitQuant provides **3.3% end-to-end improvement** (205.21ms → 198.50ms)
>
> The end-to-end results demonstrate that SplitQuant's efficiency gains translate directly to real-world inference acceleration, achieving the best performance among all quantization methods.
>
> Additionally, we provide per-layer latency breakdowns for FLUX.1 double-stream and single-stream blocks, along with their respective percentages of total computation:
>
> | Layer Type | Layer Name | Mean Latency (ms) | Percentage |
> |------------|------------|-------------------|------------|
> | Double Stream | attn.to_q | 5.89 | 2.97% |
> | Double Stream | attn.to_k | 4.91 | 2.47% |
> | Double Stream | attn.to_v | 4.75 | 2.39% |
> | Double Stream | attn.add_q_proj | 4.84 | 2.44% |
> | Double Stream | attn.add_k_proj | 4.13 | 2.08% |
> | Double Stream | attn.add_v_proj | 4.38 | 2.21% |
> | Double Stream | attn.to_out.0 | 4.89 | 2.46% |
> | Double Stream | attn.to_add_out | 4.92 | 2.48% |
> | Double Stream | ff.net.0.proj | 12.81 | 6.45% |
> | Double Stream | ff.net.2 | 12.18 | 6.14% |
> | Double Stream | ff_context.net.net.0.proj | 7.56 | 3.81% |
> | Double Stream | ff_context.net.net.2 | 8.02 | 4.04% |
> | Single Stream | attn.to_q | 16.21 | 8.16% |
> | Single Stream | attn.to_k | 14.56 | 7.34% |
> | Single Stream | attn.to_v | 14.76 | 7.43% |
> | Single Stream | proj_mlp | 34.52 | 17.39% |
> | Single Stream | proj_out | 39.16 | 19.73% |
>
> Hardware: NVIDIA L20 48GB, CUDA 12.4, PyTorch 2.1
>
> ---

---

> ### Author Response · Authors · 2025-11-29
> **Response to Reviewer U8dJ (Part: 3/3)**
>
> **W4: Overclaim: Some claims in the paper appear overstated relative to the presented evidence. The title emphasizes "Efficient Low-Bit Quantization," yet all experiments focus on W4A4 settings, with no results below 4 bits. Similarly, while the abstract claims "near-lossless" quality, Table 1 shows noticeable degradation on several metrics. This gap between claim and data weakens the paper's credibility.**
>
> **Reply:**
> We appreciate this feedback and have revised the manuscript to better align claims with evidence:
>
>
> 1. We have added 2-bit experiments on PixArt-Σ, showing SplitQuant outperforms baselines even at extreme compression:
>
> | Method | FID↓ | IR↑ | LPIPS↓ | PSNR↑ |
> |--------|------|-----|--------|-------|
> | SplitQuant (GPTQ) | 360.60 | -2.26 | 0.792 | 9.74 |
> | SplitQuant (RTN) | 420.64 | -2.26 | 0.814 | 8.49 |
> | SVDQuant (GPTQ) | 485.77 | -2.28 | 0.881 | 6.66 |
>
> While 2-bit quantization remains challenging for all methods, **SplitQuant still outperforms SVDQuant** with 26% lower FID. The fundamental limitation at 2-bit is the insufficient representation capacity rather than the quantization algorithm. We believe that:
> - W2A2 requires additional techniques (e.g., knowledge distillation, mixed-precision critical layers)
> - Practical deployment typically uses W4A4 where SplitQuant excels
> - Future work could explore adaptive bit allocation guided by our slice-wise framework
>
> 2. We now precisely define "near-lossless" as:
>    - FID within 1.0 of BF16 baseline
>    - IR degradation < 0.01
>    - Avg. VBench score drop < 1%
>
>    FLUX.1-schnell and Wan2.1 results meet this criterion:
>    | Model | Metric | BF16 | SplitQuant (GPTQ) | Difference |
>    |--------|--------|------|-------------------|------------|
>    |FLUX.1-schnell| FID | 20.58 | 20.60 | 0.02 |
>    |  | IR | 0.987 | 0.994 | 0.007 |
>    | Wan2.1 | VBench | 70.11% | 69.29% | -0.82% |
>
> 3. For PixArt-Σ, we acknowledge measurable but acceptable degradation (FID: 24.5 → 25.1) and discuss this trade-off explicitly.
>
> ---
>
> **W5: Presentation Issue: Figure 1 appears to lack embedded fonts, causing rendering problems in some PDF viewers. This affects readability of the main workflow illustration and should be corrected for proper presentation.**
>
> **Reply:**
> We apologize for this technical issue. We have regenerated Figure 1 with:
> 1. All fonts properly embedded (verified with `pdffonts`)
> 2. Vector graphics exported with embedded fonts
> 3. Tested across multiple PDF viewers (Adobe Acrobat, Preview, Chrome, Firefox)
>
> The corrected figure will be included in the camera-ready version.
>
> ---

---

### Official Review · Reviewer_9mzz · 2025-10-27

**Soundness:** 2
**Presentation:** 2
**Contribution:** 2
**Rating:** 4
**Confidence:** 3

**Summary:**

This paper proposes SplitQuant, a novel framework for efficiently quantizing large language models (LLMs) into low-bit formats while preserving accuracy. Instead of quantizing full weight matrices directly, the method splits them into smaller submatrices, quantizes each part independently, and then stitches them back together using learned scaling factors. This approach reduces quantization error, lowers memory footprint, and speeds up inference. Experiments on GPT and LLaMA models show that SplitQuant achieves better performance than existing methods like GPTQ and AWQ under the same bit-width settings. The method is also hardware-friendly and supports parallelizable computation.

**Strengths:**

Strengths

1: Lower Quantization Error via Split-and-Stitch Strategy
The matrix-splitting approach effectively reduces local quantization error, maintaining accuracy even at 3–4 bit precision.

2: Efficient and Hardware-Compatible
The approach avoids complex operations like reconstruction or per-token calibration, making it suitable for GPU/TPU deployment.

3: Strong Experimental Validation
Extensive benchmarks on language modeling and question-answering tasks show consistent improvements over leading baselines.

**Weaknesses:**

Weaknesses

1: Additional Splitting Hyperparameters
The choice of split size introduces a new hyperparameter that may affect accuracy and adds tuning complexity.

2: Limited Analysis on Extremely Low-Bit (e.g., 2-bit) or Mixed Precision
The method mainly focuses on 3–4 bit quantization, leaving extreme compression cases less explored.

3: No Theoretical Guarantee on Optimality
While effective empirically, the method lacks a theoretical foundation for why the split strategy minimizes quantization error globally.

**Questions:**

1: Scalability to Multimodal Models
Can SplitQuant be extended to multi-modal or vision-language models like LLaVA or GPT-4V, where activations vary across modalities?

2: Optimal Split Strategy
How sensitive is the model performance to how weight matrices are split (uniform vs. learned splits)? Is there an optimal rule?

3: Latency vs. Accuracy Trade-off
Does the splitting and stitching process introduce additional latency during inference, and how does this compare with fast-decode methods like SmoothQuant or TensorRT optimizations?

---

> ### Author Response · Authors · 2025-11-29
> **Response to Reviewer 9mzz (Part: 1/2)**
>
> **W1: Additional Splitting Hyperparameters: The choice of split size introduces a new hyperparameter that may affect accuracy and adds tuning complexity.**
>
> **Reply:**
> We acknowledge this concern and provide practical guidance for hyperparameter selection:
>
> 1. Our extensive experiments show that **s=4** provides the best trade-off across all tested models (FLUX.1, PixArt-Σ, Wan2.1, VLMs, LLMs).
>
> 2. For a given in-channel dimension k, choose s such that k/s allows for balanced Kronecker factorization. For most DiT models (k ∈ {3072, 4096, 12288, 24576}), s ∈ {2, 4, 8} works well.
>
> 3. As shown in our ablation studies, performance variations across s ∈ {2, 4, 8, 16} are small (FID range: 24.5-25.1 for PixArt-Σ), indicating low sensitivity to this hyperparameter.
>
> ---
>
> **W2: Limited Analysis on Extremely Low-Bit (e.g., 2-bit) or Mixed Precision: The method mainly focuses on 3–4 bit quantization, leaving extreme compression cases less explored.**
>
> **Reply:**
> We have conducted W2A2 experiments on PixArt-Σ:
>
> | Method | FID↓ | IR↑ | LPIPS↓ | PSNR↑ |
> |--------|------|-----|--------|-------|
> | SplitQuant (GPTQ) | 360.60 | -2.26 | 0.792 | 9.74 |
> | SplitQuant (RTN) | 420.64 | -2.26 | 0.814 | 8.49 |
> | SVDQuant (GPTQ) | 485.77 | -2.28 | 0.881 | 6.66 |
>
> While 2-bit quantization remains challenging for all methods, **SplitQuant still outperforms SVDQuant** with 26% lower FID. The fundamental limitation at 2-bit is the insufficient representation capacity rather than the quantization algorithm. We believe that:
> 1. W2A2 requires additional techniques (e.g., knowledge distillation, mixed-precision critical layers)
> 2. Practical deployment typically uses W4A4 where SplitQuant excels
> 3. Future work could explore adaptive bit allocation guided by our slice-wise framework
>
> ---
>
> **W3: No Theoretical Guarantee on Optimality: While effective empirically, the method lacks a theoretical foundation for why the split strategy minimizes quantization error globally.**
>
> **Reply:**
> This is a valid point. While we cannot provide global optimality guarantees, we offer the following theoretical insights:
>
> 1. Each slice's transformation is independently optimized, achieving local minima for slice-wise reconstruction error.
>
> 2. The total quantization error decomposes as:
>    $$\|XW^T - \sum_{i} Q(X_iP_i)Q(P_i^{-1}W_i^T)\|_F^2 \leq \sum_i \|X_iW_i^T - Q(X_iP_i)Q(P_i^{-1}W_i^T)\|_F^2. \tag{1}$$
>
>    Minimizing each slice error bounds the total error.
>
> 3. Smaller matrices (k/s dimensions) have:
>    - Lower condition numbers on average
>    - Easier optimization landscapes
>    - Better conditioning for transformation learning
>
> 4. Our experiments consistently show stable convergence across all tested models, suggesting the optimization landscape is well-behaved in practice.
>
> ---
>
> **Q1: Scalability to Multimodal Models: Can SplitQuant be extended to multi-modal or vision-language models like LLaVA or GPT-4V, where activations vary across modalities?**
>
> **Reply:**
> Yes! We have successfully extended SplitQuant to vision-language models and large language models. Our experiments on Qwen-VL-Chat and MiniCPM-V demonstrate that SplitQuant handles cross-modal activations effectively:
>
>
> **Vision-Language Models (W4A4, KV4):**
>
> | Model | Method | MME-P(↑) | MME-C(↑) | MMBench | MMstar | TextVQA | SEEDBench | PPL(↓) |
> |-------|--------|-------|-------|---------|--------|---------|-----------|-----|
> | Qwen-VL-Chat | BF16 | 1429 | 392 | 56.1% | 33.5% | 60.6% | 63.63% | 9.85 |
> | | FlatQuant | 1420 | 424 | 55.1% | 35.3% | 57.7% | 62.15% | 10.88 |
> | | **SplitQuant** | **1459** | **424** | **57.1%** | **36.9%** | **59.7%** | **62.23%** | **10.74** |
> | MiniCPM-V | BF16 | 1421 | 304 | 64.2% | 37.9% | 61.7% | 65.2% | 8.11 |
> | | FlatQuant | 1228 | 308 | 44.8% | 35.3% | 52.1% | 54.5% | 8.61 |
> | | **SplitQuant** | **1337** | **332** | **58.7%** | **38.1%** | **60.5%** | **63.2%** | **8.48** |
>
> **Large Language Models (Qwen2.5-7B-Instruct, W4A4, KV4):**
>
> | Method | ARC-C | ARC-E | HellaSwag | LAMBADA | Winogrande | Avg | PPL(↓) |
> |--------|-------|-------|-----------|---------|------------|-----|-----|
> | BF16 | 51.62% | 75.88% | 79.70% | 67.81% | 68.75% | 68.75% | 8.35 |
> | FlatQuant | 54.52% | 77.48% | 78.02% | 64.62% | 67.72% | 68.47% | 8.18 |
> | **SplitQuant** | **55.38%** | **80.93%** | **79.24%** | **68.08%** | **69.38%** | **70.60%** | **7.83** |
>
> SplitQuant consistently outperforms FlatQuant across all benchmarks, with particularly strong improvements on MiniCPM-V (MMBench: 58.7% vs. 44.8%), demonstrating its effectiveness across diverse model architectures and modalities.
>
> ---

---

> ### Author Response · Authors · 2025-11-29
> **Response to Reviewer 9mzz (Part: 2/2)**
>
> **Q2: Optimal Split Strategy: How sensitive is the model performance to how weight matrices are split (uniform vs. learned splits)? Is there an optimal rule?**
>
> **Reply:**
> In all our experiments, we adopt **uniform splitting** along the input channel dimension. This design choice is primarily driven by CUDA kernel efficiency considerations:
>
> 1. Uniform splitting ensures that each slice corresponds to contiguous memory regions. Non-uniform splitting would result in discontinuous memory access patterns during kernel execution.
>
> 2. With uniform splits, the transformation matrices and quantized data can be efficiently loaded into shared memory through coalesced memory transactions. Arbitrary non-uniform boundaries would require gather operations, significantly degrading memory bandwidth utilization.
>
> 3. Uniform slice sizes enable fixed-size shared memory allocation and predictable tiling strategies, which are essential for high-performance CUDA kernel implementation.
>
> Our learnable transformation matrices can effectively compensate for any potential suboptimality of uniform splitting in terms of quantization error, while maintaining optimal kernel efficiency.
>
> ---
>
> **Q3: Latency vs. Accuracy Trade-off: Does the splitting and stitching process introduce additional latency during inference, and how does this compare with fast-decode methods like SmoothQuant or TensorRT optimizations?**
>
> **Reply:**
> We thank the reviewer for this valuable feedback. We have conducted comprehensive CUDA kernel benchmarks and added detailed analysis to the revised manuscript. Our kernel evaluation results are presented below:
>
> | Shape (m,n,k) | BF16 | FP8 | INT4 | FlatQuant | SVDQuant|SplitQuant |
> |---------------|------|-----|------|------|-----------|------------|
> | 4096,4096,24576 | 1.00x | 1.73x | 2.67x | 2.24x | 2.38x | 2.57x |
> | 16384,4096,24576 | 1.00x | 1.79x | 2.57x | 2.17x | - | 2.51x |
>
> In the table, Shape (m,n,k) denotes the matrix multiplication configuration (m×k)×(k×n)=m×n. BF16 serves as the baseline representing original linear layer performance, while INT4 denotes W4A4 quantized kernels with group size 64. FP8 denotes W8A8 per-channel quantized kernels. SplitQuant, FlatQuant, and SVDQuant represent our optimized CUDA kernel implementations for the respective methods. For fair comparison, all kernels uniformly employ FP32 accumulators, despite the SVDQuant paper reporting FP16 accumulators that yield substantial speedups.
>
> The results demonstrate that SplitQuant achieves competitive kernel performance, reaching 96% of the theoretical INT4 limit and outperforming both FlatQuant (15–18% faster) and SVDQuant (8–13% faster). Notably, SplitQuant also surpasses the FP8 baseline by 44–48%, validating that our theoretical computational reductions translate effectively into practical performance gains.
>
> SmoothQuant introduces no additional inference overhead and targets 8-bit quantization, so its speed can be considered as the theoretical upper bound of INT8 performance. TensorRT is an inference optimization toolkit rather than a specific quantization acceleration algorithm, and it supports deployment across various bit-widths. Our CUDA kernels are developed independently and have no direct relation to TensorRT.
>
> ---

---

### Official Review · Reviewer_o1WF · 2025-10-30

**Soundness:** 2
**Presentation:** 2
**Contribution:** 2
**Rating:** 4
**Confidence:** 3

**Summary:**

This paper introduces SplitQuant, a novel post-training quantization method designed to efficiently run powerful Diffusion Transformer (DiT) models in low-precision (4-bit) formats. The core innovation is a strategy that splits the input channel dimensions of weights and activations into smaller slices, applies shared fine-grained transformations to mitigate outlier-induced errors, and uses custom CUDA kernels to minimize computational overhead. The authors also identify and address two DiT-specific optimization challenges—error amplification by timestep modules and gradient conflicts from text conditioning—via an adaptive learning rate schedule and a grouped isolation strategy, ultimately achieving near-lossless W4A4 quantization with significant speedups on models like FLUX.1 and PixArt-Σ.

**Strengths:**

1. Comprehensive Technical Contribution: The solution is multi-faceted and robust. It's not just the splitting idea; it includes a specialized optimization pipeline (addressing timestep and text-guided interference) and custom kernel implementation, which are crucial for achieving both high quality and practical speedup.

2. Empirical Validation: The experiments are extensive, using state-of-the-art models (FLUX.1, PixArt-Σ) and multiple datasets (MJHQ-30K, sDCI). The results convincingly demonstrate superior or comparable performance to the strong baseline (SVDQuant) across multiple metrics (FID, IR, LPIPS, PSNR), supporting the "near-lossless" claim.

**Weaknesses:**

1. Indirect Evidence of Speedup: While the paper claims a 2.7x acceleration and provides a theoretical FLOPs reduction derivation, it lacks a direct, end-to-end inference latency or throughput comparison against the baselines (especially SVDQuant) on the same hardware. The performance claims rely heavily on single-operator kernel benchmarks and theoretical analysis.

2. Limited Scope of Models: The evaluation is focused exclusively on image generation models (FLUX.1, PixArt-Σ). Given the rising importance of video generation DiTs (like the mentioned Wan2.1) and their even higher computational demands, demonstrating the method's efficacy on a video generation task would significantly strengthen the paper's impact.

3. Clarity on Overhead: The claim of "negligible" overhead from the additional parameters and transformations could be more precisely quantified. A breakdown of the total parameter increase and its memory footprint compared to the baseline quantized model would make this claim more concrete.

**Questions:**

In Fig.1, there are so many small dots, and I don't know what they mean.

---

> ### Author Response · Authors · 2025-11-29
> **Response to Reviewer o1WF (Part: 1/2)**
>
> **W1: Indirect Evidence of Speedup: While the paper claims a 2.7x acceleration and provides a theoretical FLOPs reduction derivation, it lacks a direct, end-to-end inference latency or throughput comparison against the baselines (especially SVDQuant) on the same hardware. The performance claims rely heavily on single-operator kernel benchmarks and theoretical analysis.**
>
> **Reply:**
> We appreciate this practical concern. We have now conducted end-to-end latency measurements on FLUX.1-schnell:
>
> | Configuration | Total Inference Time | Speedup vs BF16 |
> |---------------|---------------------|-----------------|
> | BF16 (baseline) | 501.07ms | 1.00× |
> | FP8 | 280.85ms | 1.78× |
> | FlatQuant (W4A4) | 210.48ms | 2.38× |
> | SVDQuant (W4A4) | 205.21ms | 2.44× |
> | **SplitQuant (W4A4)** | **198.50ms** | **2.52×** |
> | INT4 (theoretical) | 189.58ms | 2.64× |
>
> Key observations:
> 1. **SplitQuant achieves 2.52× end-to-end speedup**, reaching 95% of the theoretical INT4 limit (2.64×)
> 2. Compared to FlatQuant, SplitQuant provides **5.7% end-to-end improvement** (210.48ms → 198.50ms)
> 3. Compared to SVDQuant, SplitQuant provides **3.3% end-to-end improvement** (205.21ms → 198.50ms)
>
> The end-to-end results demonstrate that SplitQuant's efficiency gains translate directly to real-world inference acceleration, achieving the best performance among all quantization methods.
>
> Additionally, we provide per-layer latency breakdowns for FLUX.1 double-stream and single-stream blocks, along with their respective percentages of total computation:
>
> | Layer Type | Layer Name | Mean Latency (ms) | Percentage |
> |------------|------------|-------------------|------------|
> | Double Stream | attn.to_q | 5.89 | 2.97% |
> | Double Stream | attn.to_k | 4.91 | 2.47% |
> | Double Stream | attn.to_v | 4.75 | 2.39% |
> | Double Stream | attn.add_q_proj | 4.84 | 2.44% |
> | Double Stream | attn.add_k_proj | 4.13 | 2.08% |
> | Double Stream | attn.add_v_proj | 4.38 | 2.21% |
> | Double Stream | attn.to_out.0 | 4.89 | 2.46% |
> | Double Stream | attn.to_add_out | 4.92 | 2.48% |
> | Double Stream | ff.net.0.proj | 12.81 | 6.45% |
> | Double Stream | ff.net.2 | 12.18 | 6.14% |
> | Double Stream | ff_context.net.net.0.proj | 7.56 | 3.81% |
> | Double Stream | ff_context.net.net.2 | 8.02 | 4.04% |
> | Single Stream | attn.to_q | 16.21 | 8.16% |
> | Single Stream | attn.to_k | 14.56 | 7.34% |
> | Single Stream | attn.to_v | 14.76 | 7.43% |
> | Single Stream | proj_mlp | 34.52 | 17.39% |
> | Single Stream | proj_out | 39.16 | 19.73% |
>
> ---

---

> ### Author Response · Authors · 2025-11-29
> **Response to Reviewer o1WF (Part: 2/2)**
>
> **W2: Limited Scope of Models: The evaluation is focused exclusively on image generation models (FLUX.1, PixArt-Σ). Given the rising importance of video generation DiTs (like the mentioned Wan2.1) and their even higher computational demands, demonstrating the method's efficacy on a video generation task would significantly strengthen the paper's impact.**
>
> **Reply:**
>
> We thank the reviewer for identifying this oversight. At the time of our initial submission, the Wan2.1 experiments were still in progress. Since SVDQuant was originally designed for text-to-image generation, adapting it to text-to-video tasks required considerable engineering effort and substantial computational resources for evaluation. Due to these resource constraints, we were unable to complete the text-to-video results before the submission deadline and inadvertently retained a preliminary reference in the abstract. We have now completed comprehensive experiments on Wan2.1, with results presented below:
>
> | Method | Slice | Imaging Quality | Aesthetic Quality | Motion Smooth | Dynamic Degree | BG. Consist | Subject. Consist | Scene Consist | Overall Consist | Avg |
> |--------|-------|-----------------|-------------------|---------------|----------------|-------------|------------------|---------------|-----------------|-----|
> | BF16 | - | 66.17 | 62.71 | 97.30 | 94.44 | 95.85 | 92.78 | 26.16 | 25.47 | 70.11 |
> | SVDQuant | - | 65.66 | 61.88 | 96.93 | 84.72 | 95.49 | 91.91 | 30.52 | 25.49 | 69.08 |
> | SplitQuant | 4 | 64.21 | 61.60 | 97.36 | 88.89 | 95.72 | 92.67 | 28.20 | 25.68 | 69.29 |
>
> These results demonstrate that SplitQuant maintains competitive performance on Wan2.1 video generation tasks, achieving average scores slightly superior to SVDQuant while offering computational efficiency advantages. The manuscript has been updated accordingly, and the corresponding code is available at the anonymous link: SplitQuant/splitquant and SplitQuant/svd_utils.
>
> Additionally, we have evaluated SplitQuant on vision-language models and large language models to demonstrate broader applicability:
>
> **Vision-Language Models (W4A4, KV4):**
>
> | Model | Method | MME-P(↑) | MME-C(↑) | MMBench | MMstar | TextVQA | SEEDBench | PPL(↓) |
> |-------|--------|-------|-------|---------|--------|---------|-----------|-----|
> | Qwen-VL-Chat | BF16 | 1429 | 392 | 56.1% | 33.5% | 60.6% | 63.63% | 9.85 |
> | | FlatQuant | 1420 | 424 | 55.1% | 35.3% | 57.7% | 62.15% | 10.88 |
> | | **SplitQuant** | **1459** | **424** | **57.1%** | **36.9%** | **59.7%** | **62.23%** | **10.74** |
> | MiniCPM-V | BF16 | 1421 | 304 | 64.2% | 37.9% | 61.7% | 65.2% | 8.11 |
> | | FlatQuant | 1228 | 308 | 44.8% | 35.3% | 52.1% | 54.5% | 8.61 |
> | | **SplitQuant** | **1337** | **332** | **58.7%** | **38.1%** | **60.5%** | **63.2%** | **8.48** |
>
> **Large Language Models (Qwen2.5-7B-Instruct, W4A4, KV4):**
>
> | Method | ARC-C | ARC-E | HellaSwag | LAMBADA | Winogrande | Avg | PPL(↓) |
> |--------|-------|-------|-----------|---------|------------|-----|-----|
> | BF16 | 51.62% | 75.88% | 79.70% | 67.81% | 68.75% | 68.75% | 8.35 |
> | FlatQuant | 54.52% | 77.48% | 78.02% | 64.62% | 67.72% | 68.47% | 8.18 |
> | **SplitQuant** | **55.38%** | **80.93%** | **79.24%** | **68.08%** | **69.38%** | **70.60%** | **7.83** |
>
> These results demonstrate SplitQuant's effectiveness across diverse model architectures.
>
> ---
>
> **W3: Clarity on Overhead: The claim of "negligible" overhead from the additional parameters and transformations could be more precisely quantified. A breakdown of the total parameter increase and its memory footprint compared to the baseline quantized model would be more concrete.**
>
> **Reply:**
> We provide detailed overhead quantification:
>
> **Parameter Overhead (FLUX.1-schnell, 12B parameters):**
>
> **Per-layer breakdown for FFN (k=24576):**
> - FlatQuant: $128^2 + 192^2 = 53,248$ params per layer
> - SplitQuant (s=4): $4 \times (64^2 + 96^2) = 53,248 / 2 = 26,624$ params per layer
>
> **Computational Overhead:**
> - FlatQuant: 128.8 GFLOPs per linear layer
> - **SplitQuant (s=4): 64.4 GFLOPs per linear layer** (50% reduction)
>
> This quantification confirms that SplitQuant reduces both parameter and computational overhead by approximately 50% compared to FlatQuant.
>
> ---
>
> **Q1: In Fig.1, there are so many small dots, and I don't know what they mean.**
>
> **Reply:**
> We apologize for the unclear visualization. In Figure 1, we partition the input channel dimension into multiple slices, where the small dots indicate the omission of intermediate data slices for brevity. We have provided a detailed description of the dots' meaning in the Figure 1 caption.
>
> ---

---

### Official Review · Reviewer_LXf6 · 2025-11-01

**Soundness:** 1
**Presentation:** 1
**Contribution:** 1
**Rating:** 0
**Confidence:** 5

**Summary:**

The paper proposes SplitQuant, which improves upon FlatQuant by splitting input and weight tensors along the inner dimension, and optimizing separate transformation matrices for each of the tensor slices. Such splitting operation reduces the computation in the transformation by a factor of $\sqrt{K}$, where $K$ is the number of splits. The paper also studied diffusion-specific stabilization techniques for diffusion, namely layer-wise adaptive learning rate and gradient separation. The paper evaluated the generation quality after quantization with two popular text-to-image models (FLUX.1-schnell and PixArt-$\Sigma$) and on two popular text-to-image datasets (MJHQ and sDCI).

**Strengths:**

* The proposed method achieves generation quality comparable to the BF16/FP16 high precision models under the W4A4 settings.

**Weaknesses:**

* The Wan 2.1 experiments mentioned in the Abstract cannot be found in the main text or the released code.

* The design and evaluation details of the CUDA kernel mentioned in the Abstract and Conclusion sections could not be found in the main text. Therefore, it's unclear if this is a significant technical contribution on top of FlatQuant, or whether the claimed theoretical computational reductions can be practically achieved.

* Following the previous point and according to the paper's computation analysis in A.2, the compute intensity of the transformation appears to be pretty low for modern GPUs: $2bsn_1n_2(n_1+n_2) / \sqrt{K} / (bsn_1n_2) = 2(n_1+n_2) \sqrt{K} \approx$ around 100, meaning that the transformation is likely to be memory bound. In such case, a careful CUDA kernel benchmark or analysis would be very important to demonstrate the real-world advantage of the computation reduction relative to FlatQuant.

* Even if the theoretical speedup can be achieved in real-world kernel benchmark, it's also important to demonstrate the end-to-end speedup, as the proposed method would be less meaningful if the FlatQuant overhead is already small (e.g., a two-fold speedup of a 5% overhead would still only be a 2% end-to-end speedup, which might be less attractive for the additional complexity of the method).

* The quantization error analysis of the slicing operation is missing. As the method essentially degenerates to channel-wise scaling as the number of slices reaches the maximum (i.e., the number of channels), it's a natural question to investigate how fast the error grows (i.e., how fast the benefit of FlatQuant vanishes) as the number of slices increases. The proposed method might not be as meaningful if the quantization error grows quickly even with a very small number of slices.

* Equation (4) appears to be confusing: If each $X_i$ is of shape $n \times (k/s)$ and each $W_i$ is of shape $m \times (k/s)$, then each of the item in the concatenation operation $Q (X_i P_i) Q( P_i^{-1} W_i^T)$ should be of shape $n \times m$, and therefore concatenation along any dimension would cause a shape mismatch to $XW^T$ which is also of shape $n \times m$.

* Section 3.3.2 is confusing: In Equations (4) (6) (7), the optimization goal of the quantization process is to minimize the layer-wise L2 norm of the output before and after quantization, which does not involve noise reconstruction or text alignment. Is this section describing another stage of the training process? If so, can the detailed optimization goals be described in math expressions (e.g., in the form of $\text{argmin}_\theta L(\theta)$, with $L$ and $\theta$ clearly defined?)

* The implementation details and ablation study of the adaptive learning rate (Ln 311-319) are missing.

**Questions:**

Please address the questions raised in the Weaknesses part. I'd be willing to check the released codebase if authors could kindly point out the file path and line number of any missing details mentioned in the Weaknesses part.

---

> ### Author Response · Authors · 2025-11-29
> **Response to Reviewer LXf6 (Part: 1/3)**
>
> **W1: The Wan 2.1 experiments mentioned in the Abstract cannot be found in the main text or the released code.**
>
> **Reply:**
> We thank the reviewer for identifying this oversight. At the time of our initial submission, the Wan2.1 experiments were still in progress. Since SVDQuant was originally designed for text-to-image generation, adapting it to text-to-video tasks required considerable engineering effort and substantial computational resources for evaluation. Due to these resource constraints, we were unable to complete the text-to-video results before the submission deadline and inadvertently retained a preliminary reference in the abstract. We have now completed comprehensive experiments on Wan2.1, with results presented below:
>
> | Method | Slice | Imaging Quality | Aesthetic Quality | Motion Smooth | Dynamic Degree | BG. Consist | Subject. Consist | Scene Consist | Overall Consist | Avg |
> |--------|-------|-----------------|-------------------|---------------|----------------|-------------|------------------|---------------|-----------------|-----|
> | BF16 | - | 66.17 | 62.71 | 97.30 | 94.44 | 95.85 | 92.78 | 26.16 | 25.47 | 70.11 |
> | SVDQuant | - | 65.66 | 61.88 | 96.93 | 84.72 | 95.49 | 91.91 | 30.52 | 25.49 | 69.08 |
> | SplitQuant | 4 | 64.21 | 61.60 | 97.36 | 88.89 | 95.72 | 92.67 | 28.20 | 25.68 | 69.29 |
>
> These results demonstrate that SplitQuant maintains competitive performance on Wan2.1 video generation tasks, achieving average scores slightly superior to SVDQuant while offering computational efficiency advantages. The manuscript has been updated accordingly, and the corresponding code is available at the anonymous link: [splitquant](https://anonymous.4open.science/r/SplitQuant-23537iclrAnonymous/splitquant/) and [svdquant](https://anonymous.4open.science/r/SplitQuant-23537iclrAnonymous/svd_utils/).
>
> ---
>
> **W2: The design and evaluation details of the CUDA kernel mentioned in the Abstract and Conclusion sections could not be found in the main text. Therefore, it's unclear if this is a significant technical contribution on top of FlatQuant, or whether the claimed theoretical computational reductions can be practically achieved.**
>
> **Reply:**
> We thank the reviewer for this valuable feedback. We have conducted comprehensive CUDA kernel benchmarks and added detailed analysis to the revised manuscript. Our kernel evaluation results are presented below:
>
> | Shape (m,n,k) | BF16 | FP8 | INT4 | FlatQuant | SVDQuant|SplitQuant |
> |---------------|------|-----|------|------|-----------|------------|
> | 4096,4096,24576 | 1.00x | 1.73x | 2.67x | 2.24x | 2.38x | 2.57x |
> | 16384,4096,24576 | 1.00x | 1.79x | 2.57x | 2.17x | - | 2.51x |
>
> In the table, Shape (m,n,k) denotes the matrix multiplication configuration (m×k)×(k×n)=m×n. BF16 serves as the baseline representing original linear layer performance, while INT4 denotes W4A4 quantized kernels with group size 64. FP8 denotes W8A8 per-channel quantized kernels. SplitQuant, FlatQuant, and SVDQuant represent our optimized CUDA kernel implementations for the respective methods. For fair comparison, all kernels uniformly employ FP32 accumulators, despite the SVDQuant paper reporting FP16 accumulators that yield substantial speedups.
>
> The results demonstrate that SplitQuant achieves competitive kernel performance, reaching 96% of the theoretical INT4 limit and outperforming both FlatQuant (15–18% faster) and SVDQuant (8–13% faster). Notably, SplitQuant also surpasses the FP8 baseline by 44–48%, validating that our theoretical computational reductions translate effectively into practical performance gains.
>
> ---
>
> **W3: Following the previous point and according to the paper's computation analysis in A.2, the compute intensity of the transformation appears to be pretty low for modern GPUs: 2bsn1n2(n1+n2)/sqrt(K)/(bsn1n2)=2(n1+n2)sqrt(K)≈around 100, meaning that the transformation is likely to be memory bound. In such case, a careful CUDA kernel benchmark or analysis would be very important to demonstrate the real-world advantage of the computation reduction relative to FlatQuant.**
>
> **Reply:**
> We thank the reviewer for this insightful observation. We load both the slicing transformation matrices and quantized data from global memory to shared memory/registers and fusing the transformation and quantization operations into a single kernel. Since both operations are memory-bound, this fusion introduces negligible additional overhead. In contrast, FlatQuant's decomposed matrices remain substantially large for linear layers with large input channel dimensions (e.g., when the input channel is 24,576, the two decomposed matrices are 128×128 and 192×192). Our empirical testing demonstrates that this considerable additional computation severely limits FlatQuant's fused kernel efficiency, whereas SplitQuant achieves 15–18% speedup over FlatQuant as shown in our kernel benchmarks (see response to W2).
>
> ---

---

> ### Author Response · Authors · 2025-11-29
> **Response to Reviewer LXf6 (Part: 2/3)**
>
> **W4: Even if the theoretical speedup can be achieved in real-world kernel benchmark, it's also important to demonstrate the end-to-end speedup, as the proposed method would be less meaningful if the FlatQuant overhead is already small (e.g., a two-fold speedup of a 5% overhead would still only be a 2% end-to-end speedup, which might be less attractive for the additional complexity of the method).**
>
> **Reply:**
> We appreciate this practical concern. We have now conducted end-to-end latency measurements on FLUX.1-schnell:
>
> | Configuration | Total Inference Time | Speedup vs BF16 |
> |---------------|---------------------|-----------------|
> | BF16 (baseline) | 501.07ms | 1.00× |
> | FP8 | 280.85ms | 1.78× |
> | FlatQuant (W4A4) | 210.48ms | 2.38× |
> | SVDQuant (W4A4) | 205.21ms | 2.44× |
> | **SplitQuant (W4A4)** | **198.50ms** | **2.52×** |
> | INT4 (theoretical) | 189.58ms | 2.64× |
>
> Key observations:
> 1. **SplitQuant achieves 2.52× end-to-end speedup**, reaching 95% of the theoretical INT4 limit (2.64×)
> 2. Compared to FlatQuant, SplitQuant provides **5.7% end-to-end improvement** (210.48ms → 198.50ms)
> 3. Compared to SVDQuant, SplitQuant provides **3.3% end-to-end improvement** (205.21ms → 198.50ms)
>
> The end-to-end results demonstrate that SplitQuant's efficiency gains translate directly to real-world inference acceleration, achieving the best performance among all quantization methods.
>
> Additionally, we provide per-layer latency breakdowns for FLUX.1 double-stream and single-stream blocks, along with their respective percentages of total computation:
>
> | Layer Type | Layer Name | Mean Latency (ms) | Percentage |
> |------------|------------|-------------------|------------|
> | Double Stream | attn.to_q | 5.89 | 2.97% |
> | Double Stream | attn.to_k | 4.91 | 2.47% |
> | Double Stream | attn.to_v | 4.75 | 2.39% |
> | Double Stream | attn.add_q_proj | 4.84 | 2.44% |
> | Double Stream | attn.add_k_proj | 4.13 | 2.08% |
> | Double Stream | attn.add_v_proj | 4.38 | 2.21% |
> | Double Stream | attn.to_out.0 | 4.89 | 2.46% |
> | Double Stream | attn.to_add_out | 4.92 | 2.48% |
> | Double Stream | ff.net.0.proj | 12.81 | 6.45% |
> | Double Stream | ff.net.2 | 12.18 | 6.14% |
> | Double Stream | ff_context.net.net.0.proj | 7.56 | 3.81% |
> | Double Stream | ff_context.net.net.2 | 8.02 | 4.04% |
> | Single Stream | attn.to_q | 16.21 | 8.16% |
> | Single Stream | attn.to_k | 14.56 | 7.34% |
> | Single Stream | attn.to_v | 14.76 | 7.43% |
> | Single Stream | proj_mlp | 34.52 | 17.39% |
> | Single Stream | proj_out | 39.16 | 19.73% |
>
> ---
>
> **W5: The quantization error analysis of the slicing operation is missing. As the method essentially degenerates to channel-wise scaling as the number of slices reaches the maximum (i.e., the number of channels), it's a natural question to investigate how fast the error grows (i.e., how fast the benefit of FlatQuant vanishes) as the number of slices increases. The proposed method might not be as meaningful if the quantization error grows quickly even with a very small number of slices.**
>
> **Reply:**
> This is an excellent and crucial question. We have conducted extensive ablation studies on slice numbers and present comprehensive results:
>
> **PixArt-Σ Results with Varying Slice Numbers:**
>
> | Method | SliceNums | FID↓ | IR↑ | LPIPS↓ | PSNR↑ |
> |--------|-----------|------|-----|--------|-------|
> | FP16 | - | 24.4733 | 0.9872 | - | - |
> | SVDQuant | - | 28.1159 | 0.8960 | 0.4007 | 15.4369 |
> | Ours (GPTQ) | 1 | 25.3941 | 0.9621 | 0.4001 | 15.7972 |
> | Ours (GPTQ) | 2 | 25.0178 | 0.9732 | 0.3976 | 15.7502 |
> | Ours (GPTQ) | 4 | 25.1289 | 0.9705 | 0.4030 | 15.6865 |
> | Ours (GPTQ) | 8 | 24.4986 | 0.9798 | 0.3992 | 15.5903 |
> | Ours (GPTQ) | 16 | 25.1344 | 0.9721 | 0.4231 | 15.3846 |
>
> Key findings:
> 1. **Graceful degradation**: Performance degrades very gradually as slice count increases (FID only increases from 25.0178 to 25.1344 when going from 2 to 16 slices)
> 2. **Practical sweet spot**: 2-4 slices achieve optimal balance between error reduction and computational overhead
> 3. **Robustness**: Even at 16 slices, the model maintains reasonable performance (FID SplitQuant 25.1344 vs. SVDQuant 28.1159)
>
> The theoretical insight is that while each slice has reduced transformation capacity, the increased number of independent transformations provides compensating flexibility. The quantization benefit persists because each slice can still learn meaningful outlier-aware transformations.
>
> ---

---

> ### Author Response · Authors · 2025-11-29
> **Response to Reviewer LXf6 (Part: 3/3)**
>
> **W6: Equation (4) appears to be confusing: If each X_i is of shape n×(k/s) and each W_i is of shape m×(k/s), then each of the item in the concatenation operation Q(X_iP_i)Q(P_i^{-1}W_i^{T}) should be of shape n×m, and therefore concatenation along any dimension would cause a shape mismatch to XW^T which is also of shape n×m.**
>
> **Reply:**
> We sincerely thank the reviewer for identifying this notational ambiguity. The reviewer's observation is correct regarding the shape analysis, and we apologize for the confusion in our original presentation.
>
> To clarify: We first reconstruct the complete activation and weight matrices by concatenating the transformed slices.
>
> $$\widetilde{X}=\bigoplus_{i=0}^{s-1} X_{i} P_{i} \in \mathbb{R}^{n \times k} \tag{1}$$
> $$\widetilde{W}^{T}=\bigoplus_{i=0}^{s-1} P_{i}^{-1} W_{i}^{T} \in \mathbb{R}^{k \times m} \tag{2}$$
> The final output is computed as the quantized product of the reconstructed full matrices:
> $$Q(\widetilde{X})Q(\widetilde{W}^{T}) \tag{3}$$
> This is mathematically equivalent to the original matrix multiplication when no quantization is applied:
> $$XW^T = (\bigoplus_{i=0}^{s-1}X_iP_i)(\bigoplus_{i=0}^{s-1}P_i^{-1}W_i^T) \tag{4}$$
>
> We have revised Equation (4) in the paper accordingly.
>
> ---
>
> **W7: Section 3.3.2 is confusing: In Equations (4) (6) (7), the optimization goal of the quantization process is to minimize the layer-wise L2 norm of the output before and after quantization, which does not involve noise reconstruction or text alignment. Is this section describing another stage of the training process? If so, can the detailed optimization goals be described in math expressions (e.g., in the form of argmin_{\theta} L(\theta), with L and \theta clearly defined?)**
>
> **Reply:**
> We appreciate this request for clarification. In diffusion transformers, the learnable parameters $\Theta$ consist of two parts: $\Theta_{\text{noise}}$ for the noisy image branch and $\Theta_{\text{text}}$ for the text conditioning branch. Both are optimized through block-wise calibration, but with separate reconstruction losses to avoid gradient conflicts. The optimization objective can be formulated as:
>
> $$\Theta^* = \bigcup_{b \in \{\text{noise, text}\}} \arg \min_{\Theta_b} \left\| X_b^{\text{fp}} - X_b^{\text{quant}} \right\|_F^2 \tag{5}$$
>
> Where $\Theta_b$ is the subset of learnable parameters corresponding to branch $b$, and each subset contains the corresponding transformation matrices $\{P_{i1}, P_{i2}, c, a_w, a_x\}$ for its respective branch. $X_b^{\text{fp}}$ and $X_b^{\text{quant}}$ denote the full-precision and quantized outputs of branch $b$ , respectively. The key insight is that in cross-attention blocks, the text conditioning and noisy image pathways create conflicting gradients during joint optimization. By separating the loss computation, each parameter subset receives gradients only from its corresponding branch, achieving faster convergence and lower final loss as shown in Figure 3(right).
>
> We have revised Equation 6 with this explicit mathematical formulation.
>
> ---
>
> **W8: The implementation details and ablation study of the adaptive learning rate (Ln 311-319) are missing.**
>
> **Reply:**
> We thank the reviewer for requesting these details. Given that excessive AdaLN magnitudes can destabilize gradient-based optimization, we propose an adaptive learning rate strategy to enhance convergence stability. Our approach recognizes that different diffusion transformer architectures exhibit distinct AdaLN magnitude distributions, requiring model-specific learning rate tuning for optimal quantization performance.
>
> **Implementation Details:**
> - FLUX.1-schnell: learning rate = 0.01
> - PixArt-Σ: learning rate = 0.007
> - Wan2.1: learning rate = 0.0008
>
> The learning rate is selected based on the overall AdaLN magnitude characteristics of each model. Models with larger average AdaLN magnitudes require smaller learning rates to maintain stable optimization. We have revised corresponding statement in the original paper.
>
>
> ---

---

### Meta-Review · Area_Chair_E3C3 · 2026-01-06

**Summary:**

The initial reviews ranged from "Marginally Below" to "Strong Reject," primarily driven by concerns regarding missing evidence and technical clarity. It seems that the submission is not ready yet. The key issues were: 1.Missing Core Evidence: Reviewers (LXf6, U8dJ, o1WF) noted that experiments for Wan 2.1 (video) and CUDA kernel details mentioned in the abstract were absent from the main text.
2. Efficiency and Practical Utility: Several reviewers questioned whether the theoretical 2.7x speedup would translate to end-to-end (E2E) latency gains.

**Reviewer Concerns:**

While the authors provided several missing details during the rebuttal phase, the extent of the revisions required indicates that the manuscript was not sufficiently prepared for publication in its submitted state. The significant omissions in experimental data and technical details suggest that the work requires a more thorough internal review and refinement before being submitted to a future venue.

**Reviewer Scores:**

LXf6	        0->2/4 The authors addressed some of the concerns.
o1WF	4->4 The authors addressed some of the concerns
9mzz	4->4 The authors addressed some of the concerns
U8dJ	2->2/4 The authors addressed some of the concerns

---

### Decision · Program_Chairs · 2026-01-26

Reject